# *"What Is The Performance Ceiling of My Classifier?"* Utilizing Category-Wise Influence Functions for Pareto Frontier Analysis

## Abstract

Data-centric learning seeks to improve model performance from the perspective of data quality, and has been drawing increasing attention in the machine learning community. Among its key tools, influence functions provide a powerful framework to quantify the impact of individual training samples on model predictions, enabling practitioners to identify detrimental samples and retrain models on a cleaner dataset for improved performance. However, most existing work focuses on the question "what data benefits the learning model?" In this paper, we take a step further and investigate a more fundamental question: "what is the performance ceiling of the learning model?" Unlike prior studies that primarily measure improvement through overall accuracy, we emphasize category-wise accuracy and aim for Pareto improvements, ensuring that every class benefits, rather than allowing tradeoffs where some classes improve at the expense of others. To address this challenge, we propose category-wise influence functions and introduce an influence vector that quantifies the impact of each training sample across all categories. Leveraging these influence vectors, we develop a principled criterion to determine whether a model can still be improved, and further design a linear programming–based sample reweighting framework to achieve Pareto performance improvements. Through extensive experiments on synthetic datasets, vision, and text benchmarks, we demonstrate the effectiveness of our approach in estimating and achieving a model's performance improvement across multiple categories of interest.

## 1 Introduction

Data-centric learning has recently emerged as a central research topic in the machine learning community (Feldman & Zhang, 2020; Chhabra et al., 2024; Richardson et al., 2023), shifting attention from purely algorithmic model design towards improving the quality of training data. Unlike conventional preprocessing techniques such as normalization or outlier removal, that operate independently of the learning algorithm, data-centric learning is tightly coupled with the downstream model. Its primary goal is to assess whether each training sample is beneficial or detrimental with respect to a specific learning objective, thereby guiding principled data curation and optimization.

Sample influence estimation is a fundamental task in data-centric learning. A straightforward approach for estimating influence involves training the model twice– once with and once without a target sample, and then comparing the change in model performance on a validation set. However, this approach is computationally expensive and often infeasible for large-scale datasets/models. To address this challenge, the seminal work of Koh & Liang (2017) introduced *influence functions*, a technique from robust statistics (Hampel, 1974; Cook & Weisberg, 1982) that enables approximate influence estimation without retraining. This approach allows efficient estimation of sample influence, inspiring a series of follow-up studies and variants on its applications on deep models (Schioppa et al., 2022; Chhabra et al., 2025; Kwon et al., 2024; Wang et al., 2025).

Broadly speaking, sample influence estimation seeks to answer the question: *"which training samples help the model, and which ones harm it* (Chhabra et al., 2024)?" In practice, samples identified as detrimental are removed, and the model is retrained on the refined dataset, often resulting in mea-

surable performance gains. This naturally raises deeper questions: *can the model's performance be improved even further by iteratively repeating this process? And if so, what is the ultimate performance ceiling of the learning model?*

In this paper, we focus on the multi-class classification setting: given a training set, a validation set, and a classification model, we aim to answer two key questions from the data perspective: (1) has the classifier already reached its maximum potential performance, and (2) if not, how can its performance be further improved? Importantly, the notion of "improvement" in our work does not simply refer to an increase in overall accuracy. Instead, we adopt a *Pareto improvement* perspective: we seek performance gains where every class benefits, avoiding tradeoff scenarios in which some classes improve at the expense of others. We summarize our contributions as follows:

- We tackle a fundamental yet largely overlooked question: "*What is the performance ceiling of a classifier?*" Specifically, we determine whether a given classifier has already reached its maximum potential performance and, if not, how it can be further improved.

- We introduce category-wise[1] influence functions to assess the model's Pareto frontier on each category and systematically analyze its performance ceiling. Leveraging these influence scores, we further propose a linear programming–based sample reweighting framework to achieve Pareto performance improvements across classes of interest.

- We validate our category-wise influence functions on synthetic and benchmark datasets and present detailed case studies showing how to determine whether a classifier has reached its performance ceiling using our linear programming–based sample reweighting framework.

## 2 RELATED WORK

**Sample Influence Estimation.** Influence functions comprise a set of methods from robust statistics (Hampel, 1974; Cook & Weisberg, 1982) that can be used to approximately estimate influence without requiring retraining, i.e., they can help create a conceptual link that traces model performance to samples in the training set. For gradient-based models trained using empirical risk minimization, the seminal work by Koh & Liang (2017) utilizes a Taylor-series approximation and LiSSA optimization (Agarwal et al., 2017) to compute sample influences and relies on the Hessian matrix. Follow-up works such as Representer Point (Yeh et al., 2018) and Hydra (Chen et al., 2021) improve influence estimation performance for deep learning models. More recently, efficient influence estimation methods such as DataInf (Kwon et al., 2024), Arnoldi iteration (Schioppa et al., 2022), and Kronecker-factored approximation curvature (Grosse et al., 2023) have been proposed which can even be employed for larger models. Some other approaches directly utilize the gradient space to measure influence (Pruthi et al., 2020; Charpiat et al., 2019), while others use some ensemble methods (Bae et al., 2024; Kim et al., 2024; Dai & Gifford, 2023). Recent work has also found that *self-influence* only on the training set can be a useful measure for detecting sample influence (Bejan et al., 2023; Thakkar et al., 2023). Influence functions have been widely used in the community for a number of data-centric applications (Feldman & Zhang, 2020; Chhabra et al., 2024; Richardson et al., 2023), but the focus of these works has predominantly been on using the overall accuracy of the model as a proxy for performance measurement. This contrasts with the main motivation in our paper, where we seek to study how different samples in the training set influence different categories/classes of the data. Such class-wise tradeoff analysis based on the Pareto frontier is of paramount importance in multiple applications, such as category-aware domain adaptation (Xiao et al., 2024) and fair classification (Lees et al., 2019; Martinez et al., 2020).

**Pareto frontier Analysis.** Pareto frontier analysis is widely employed in many domains, where multiple objectives need to be optimized simultaneously, necessitating solutions that can effectively measure the tradeoffs between each of the objectives. For instance, Edelman et al. (2023) analyzed Pareto tradeoffs across resources for model training including data, model architectures, and computation. Lin et al. (2019) introduced Pareto multi-task learning, where multiple Pareto-optimal solutions are generated efficiently to help practitioners choose the best one according to their tradeoff preferences. Other work, such as the Iterated Pareto Referent Optimization method proposed by Röpke et al. (2025) for multi-objective reinforcement learning, decomposes the search for optimal paths into a sequence of single objectives. Similarly, Cai et al. (2023) handled distributional pareto-optimal policies in reinforcement learning under uncertainty. Pareto-optimal tradeoffs have

---

[1]We use the terms *category* and *class* interchangeably throughout the paper.

also been studied in other problem domains, such as neural architecture search (Elsken et al., 2019). As is evident, none of these works investigate category-wise tradeoffs during training and analysis of the Pareto-front for assessing the performance ceiling of a given classifier, unlike our work.

**Other Data-Centric Learning.** Many works in data-centric learning study research questions beyond Pareto frontier analysis and category-wise influence estimation. *Datamodels* (Ilyas et al., 2022) estimate training sample contributions as well, but only for one test sample at a time. Other approaches such as (Jain et al., 2023; Paul et al., 2021; Killamsetty et al., 2021) aim to accelerate deep learning training time via subset/coreset selection. Data pruning, augmentation, and relabeling approaches (Yang et al., 2022; Tan et al., 2024; Kong et al., 2021; Chhabra et al., 2022; Richardson et al., 2023) and model pruning approaches (Lyu et al., 2023) based on influence analysis have also been proposed. Another related area of research is *active learning* (Cohn et al., 1996), which seeks to iteratively identify optimal samples to annotate given a large unlabeled training data pool (Liu et al., 2021; Nguyen et al., 2022; Wei et al., 2015).

## 3 PROPOSED APPROACH

### 3.1 PRELIMINARIES

Let $T=\{z_i\}_{i=1}^n$ be a training set, where $z_i =(x_i,y_i)$ includes the input space sample features $x_i$ and output space label $y_i$. A classifier trained using empirical risk minimization on the empirical loss $\ell$ can be written as: $\hat{\theta}=\arg\min_{\theta\in\Theta} \frac{1}{n}\sum_{i=1}^n \ell(z_i;\theta)$. Influence functions (Koh & Liang, 2017) constitute methods from robust statistics (Hampel, 1974; Cook & Weisberg, 1982; Martin & Yohai, 1986) that can help measure the effect of changing an infinitesimal weight of training samples on the model utility/performance. Downweighting a training sample $z_j$ by a very small fraction $\epsilon$ leads to a model parameter: $\hat{\theta}(z_j; -\epsilon) = \arg\min_{\theta\in\Theta} \frac{1}{n}(\sum_{i=1}^n \ell(z_i;\theta)-\epsilon\ell(z_j;\theta))$. By evaluating the limit as $\epsilon$ approaches 1, we can estimate the *influence score* associated with the removal of $z_j$ from the training set in terms of loss on the validation $V$ set, without undertaking any computationally expensive leave-one-out re-training as:

$$\mathcal{I}^{\hat{\theta}}(z_j, V) = \sum_{z\in V} \nabla_{\hat{\theta}}\ell(z;\hat{\theta})^\top \mathbf{H}_{\hat{\theta}}^{-1}\nabla_{\hat{\theta}}\ell(z_j;\hat{\theta}). \tag{1}$$

where $\nabla_{\hat{\theta}}\ell(z_j;\hat{\theta})$ is the gradient of sample $z_j$ to model parameters, and $\mathbf{H}_{\hat{\theta}}=\sum_{i=1}^n \nabla_{\hat{\theta}}^2\ell(z_i;\hat{\theta})$ denotes the Hessian. Higher values of $\mathcal{I}^{\hat{\theta}}(z_j, V)$ indicate a more positively influential sample (i.e., one that decreases the overall classification loss) and conversely, lower values correspond to a more negatively influential sample. Also note that while influence functions have demonstrated their benefits in deep learning non-convex models, such as on BERT (Han et al., 2020), ResNets (Liu et al., 2021; Yang et al., 2022), and CNNs (Koh & Liang, 2017; Schioppa et al., 2022), there is ongoing research that studies their suitability to these models (Bae et al., 2022; Basu et al., 2020; Epifano et al., 2023; Schioppa et al., 2024). While this research question regarding applicability is not the focus of our paper we employ influence function formulations that have been shown to work well for deep models (Grosse et al., 2023; Kwon et al., 2024).

### 3.2 RESEARCH QUESTION

While measuring the influence of training samples on the predictive performance of a model can serve as a powerful tool for numerous data-centric learning applications, prior work (Koh & Liang, 2017; Chhabra et al., 2024; Han et al., 2020; Kwon et al., 2024; Schioppa et al., 2024; Liu et al., 2021; Yang et al., 2022; Chhabra et al., 2025) has undertaken this analysis using only the *overall accuracy* of the model as an indicator for performance. However, this is a restrictive scenario, and it can be beneficial for users/developers to consider a category-wise analysis for fine-grained understanding model's performance.

The basis for category-wise analysis stems from the fact that different training samples can lead the model to learn different predictive patterns, and hence, they can impact their own class and other classes in varying ways. Following this rationale, it should be possible to ascertain how different samples in the training set impact different categories of the dataset and therefore, utilize this information as a model developer to make relevant tradeoffs that are desirable for the

given application at hand. These tradeoffs appear in numerous learning problems– such as *fair classification* (Lees et al., 2019; Martinez et al., 2020), where performance of multiple different categories need to be maximized jointly and *category-aware active domain adaptation* (Xiao et al., 2024), where performance impacts across different categories/classes need to be individually identified, among several others. In contrast to past work which focuses solely on overall class accuracy, this forms the main motivation for our research focus in this work, where we seek to study how performance can be improved for certain classes by potentially sacrificing performance of others. The space of solutions that describe these different tradeoffs between categories is also known as the *Pareto frontier* (Lotov & Miettinen, 2008).

In this paper, we consider whether a given classifier has already reached its maximum potential performance, i.e., Pareto frontier, if not, how it can be further improved. We will now discuss our proposed methods for obtaining the Pareto frontier (and thus, the classifier's performance ceiling).

### 3.3 CATEGORY-WISE INFLUENCE VECTOR ESTIMATION

We now discuss our proposed approach for obtaining the Pareto frontier for the classifier across different categories. Analytically, the Pareto frontier for a training sample can be described using an *influence vector* $P(z) \in \mathbb{R}^K$ of length $K$ (given $K$ classes/categories) where each cell $P^k(z)$ for $k \in [K]$ is the impact to the category $k$ when the sample $z$ is removed from the training set. Next, we will develop category-wise influence scores for obtaining this Pareto frontier vector.

Let the subset of samples of set $S$ that belong to class $k$ be denoted as $S^k$. We can then measure the influence score for training sample $z$ as $P^k(z) = \mathcal{I}^{\hat{\theta}}(z, S^k)$ for all $k \in [K]$ using Eq. (1). It is important to note that the category-wise influence vector $P(z)$ is a useful solution aimed at answering the research question (*what is the classifier's performance ceiling?*) we had formulated above. More specifically, the influence vector allows users/developers to easily gauge the classifier's performance ceiling– if all values of $P(z) > 0$, the sample $z$ is *beneficial* to all categories and if all values of $P(z) \leq 0$, the sample $z$ is *detrimental* to all categories. The third case is when $P(z)$ takes on *mixed* values that are both positive and negative.

Without loss of generalizability, consider the case scenario with two categories: $\mathcal{C}_1$ and $\mathcal{C}_2$. Given a model, we can calculate the influence vector for all training samples, and visualize the influence vector in Figure 1. Training samples located in the joint negative region are expected to hinder performance in both categories; thus, their removal could lead to simultaneous improvements in both aspects. Conversely, samples in the joint positive region can be further leveraged or retrained to enhance performance. These observations imply that the presence of samples in either the joint negative or positive regions indicates room for Pareto improvement, suggesting that the current model has not yet reached the Pareto frontier. However, an important question arises: *if no samples exist in the joint negative or positive regions, does this imply that the Pareto frontier has been achieved?* This question motivates further investigation into the geometric conditions and theoretical guarantees for achieving the Pareto frontier.

The answer to the above question is no, indicating that the current model can still be improved. At the individual sample level, if all samples may be located within tradeoff regions, involving or removing any sample will lead the tradeoff effect. However, when samples are considered collectively as a set, combining certain samples can yield a new sample that falls into the joint positive region. For example, as shown by the red arrows $z_1$ and $z_2$ in Figure 1, these two training samples lie in different tradeoff regions. Yet, when combined, the new sample becomes jointly beneficial to both categories. This observation suggests that a reweighting strategy

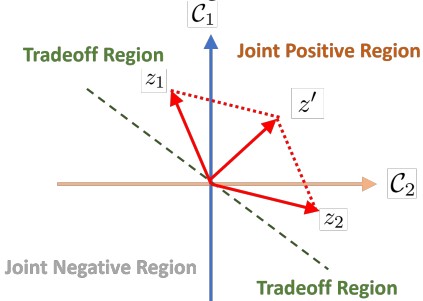

Figure 1: Influence space for 2 categories.

*can* be employed to modify the training set for Pareto improvements. Unlike approaches that simply decide whether to include or exclude individual samples, reweighting considers combinations of samples, providing a more general and flexible mechanism for optimization. Based on this insight, we can establish a new condition for achieving the Pareto frontier: all samples are close to the line $y = -x$ (the dashed green line). In the following, we will provide a reweighting framework based on the influence vector to achieve Pareto improvement.

---

**Algorithm 1** PARETO-LP-GA

---

**Input:** Training set $T$, model parameters $\hat{\theta}^e$ for epoch $e$, influence vector $P(z), \forall z \in T$, target set $\mathcal{C}_{\text{target}}$, GA iterations $G$

**Output**: Optimized per-sample weights $w^*$, optimized class-wise performance thresholds $\alpha^*$

1: **initialize** population $\alpha^0 = \{\alpha_k^0\}_{k \in [K]}$ randomly
2: **for** $g \in [G]$ **do**
3:      **for** candidate threshold set $\alpha^g$ in current population **do**
4:      **solve** the following LP to obtain $w$:

$$\max_{w} \quad \sum_{k \in \mathcal{C}_{\text{target}}} \sum_{z_i \in T} w_i P^k(z_i)$$

$$\text{subject to} \quad \sum_{z_i \in T} w_i P^k(z_i) \geq \alpha_k^g \sum_{z_i \in T} P^k(z_i), \quad \forall k \in [K]$$

5:      $\hat{\theta}^{e+1} \leftarrow \text{TRAINONEEPOCH}(\hat{\theta}^e, T, w)$
6:      **compute** relative change in performance $\Delta_k^{e+1}$ from $\hat{\theta}^e$ to $\hat{\theta}^{e+1}$ for $k \in [K]$
7:      **compute** fitness $F(\alpha^g)$ for current candidate threshold set as follows:

$$F(\alpha^g) = \frac{1}{|\mathcal{C}_{\text{target}}|} \sum_{k \in \mathcal{C}_{\text{target}}} \mathbb{1}_{[\Delta_k^{e+1} \leq 0]}(-\infty) + \frac{1}{|\mathcal{C} \setminus \mathcal{C}_{\text{target}}|} \sum_{k \notin \mathcal{C}_{\text{target}}} \mathbb{1}_{[\Delta_k^{e+1} < 0]}(\Delta_k^{e+1})$$

8:      **end for**
9:      **apply** selection, crossover, and mutation operations on population
10:      **store** $\alpha^*$ and $w^*$ that maximizes fitness so far
11: **end for**
12: **return** $w^*, \alpha^*$

---

### 3.4 PARETO-LP-GA: IMPROVING PARETO PERFORMANCE USING INFLUENCE VECTORS

We now aim to utilize our category-aware influence vector $P(z), \forall z \in T$ to improve the performance of a given classifier. More specifically, we utilize the category-wise influence vector to obtain per-sample weights for training losses that improve performance on a target subset of categories while controlling degradation on the remaining categories, via *linear programming* (LP) (Dantzig, 2002). Furthermore, while influence scores are useful estimators for tuning category-wise performance, they cannot be utilized to obtain class-wise performance thresholds (equivalently, *slack variables* in the linear program). Hence, we utilize a *genetic algorithm* (GA) (Forrest, 1996) to help identify these class-specific slack variables and ensure that the weighted LP obtains highly optimized solutions. We thus propose our approach PARETO-LP-GA, which is an influence vector-guided linear programming approach for training sample weight optimization combined with a GA search.

We apply this weighted model in the context of two different settings: (a) *Direct Improvement (DI)*: this refers to improving specific categories in a particular epoch as desired by the model developer where target categories are selected by the developer based on current per-class accuracy observations; and (b) *Course Correction (CC)*: the developer while training the model observes accuracy drops in a certain epoch, and decides to modify the training trajectory for the detrimental epoch identified. A detrimental epoch means one after which the accuracy of some classes decreases significantly, indicating potential Pareto-optimal class/category tradeoffs.

The PARETO-LP-GA procedure is provided in Algorithm 1. Denoting the full set of $K$ classes as $\mathcal{C}$, let $\mathcal{C}_{\text{target}}$ be the target subset of categories we wish to improve the performance for, while ensuring minimal performance degradation in other classes $\mathcal{C} \setminus \mathcal{C}_{\text{target}}$. Algorithm 1 takes in as input the training set $T$, the model parameters trained until a certain epoch $e$, our category-aware influence vector $P(z)$, the target classes $\mathcal{C}_{\text{target}}$ to improve performance for, the total iterations $G$. First, we initialize the population variable that controls for class-wise performance threshold along the Pareto frontier randomly, denoted as $\alpha_k, \forall k \in [K]$. Then, for each iteration of the genetic algorithm (GA), we solve a linear program (Line 4) that seeks to optimize for the per-sample weights by ensuring estimated performance via the category-aware influence vector is maximized on target classes while ensuring class-wise performance is above each category/class threshold ($\alpha_k$). Subsequently, we train the model for the next epoch ($e + 1$) by applying the current optimized weight set (Line 5). We then

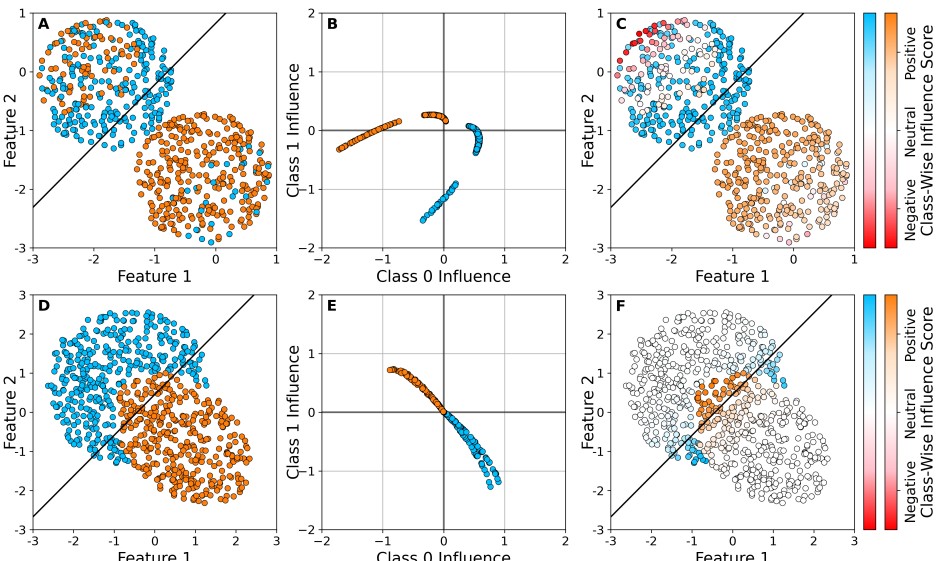

Figure 2: Validation of our category-wise influence function methods for analyzing the Pareto frontier on two synthetic binary classification datasets with logistic regression. Subfigures **A-C** showcase results on a synthetic dataset that is linearly separable and contains noisy detrimental training samples, where performance can improve by mislabeled sample removal. Subfigures **D-F** detail results for our method on a non-linearly separable dataset without any noisy samples, where performance improvements cannot be made for either class without sacrificing performance for the other. Subfigures **A** and **D** showcase the distribution of training samples for each of the two datasets with blue and orange denoting the ground-truth class labels. Subfigures **B** and **E** showcase the category-wise influence score distribution for both datasets. Further, subfigures **C** and **F** map the influence values to the training samples using color intensity in accordance with class colors to denote the influence magnitudes, where the original class color means positive and red color means negative.

measure relative change in performance ($\Delta_k^{e+1}$) between epochs $e \to e + 1$ for *Direct Improvement* (for *Course Correction* this performance change $\Delta_k^{e+1}$ is instead calculated between the original epoch $e + 1$ and the newly weighted epoch $e + 1$; the rest of the procedure remains identical). Now, while we have optimized the weight set, we still need to obtain the optimal class-wise thresholds via the GA search. Hence, we formulate the fitness function (Line 7) such that if performance for $\mathcal{C}_{\text{target}}$ decreases at all (i.e., $\Delta_k^{e+1} \leq 0$) the fitness is set to a large-magnitude negative value (denoted as $-\infty$). Moreover, for non-target classes $\mathcal{C} \setminus \mathcal{C}_{\text{target}}$, if performance decreases, the fitness score reflects the degree of degradation. Thus, the GA $\alpha$ search also optimizes for performance improvement along desired target classes while ensuring minimal performance reduction across non-target classes. Post this step, we apply the standard GA operations (selection, crossover, mutation, etc.) on the population. Eventually, the algorithm return the optimized weight set $w^* = [w_1^*, \ldots, w_n^*]$ for the training set, that will be applied to the loss computation during next epoch training.

## 4 SYNTHETIC DATA VERIFICATION

We now analyze the efficacy of our criterion on the model's performance ceiling with two synthetic binary classification datasets using logistic regression, as shown in Figure 2. The top row subfigures **A-C** denote a linearly separable dataset which consists of noisy samples that are mislabeled. The dataset consists of 300 blue class samples and 300 orange class samples,[2] generated using a circular uniform distribution. Noises were added to the training set by choosing random points from each group, 50 from blue and 20 from orange, and then flipping their label. Clearly, removing these noisy samples should improve performance and our category-wise influence functions should reflect their detrimental influence for both classes/categories. The large majority of non-noisy samples should positively influence one of the classes and negatively influence the other class. Hence, removing these samples should sacrifice performance for one of the classes and should be reflected in the

---

[2]The descriptions of data and models can be seen in Appendix A.

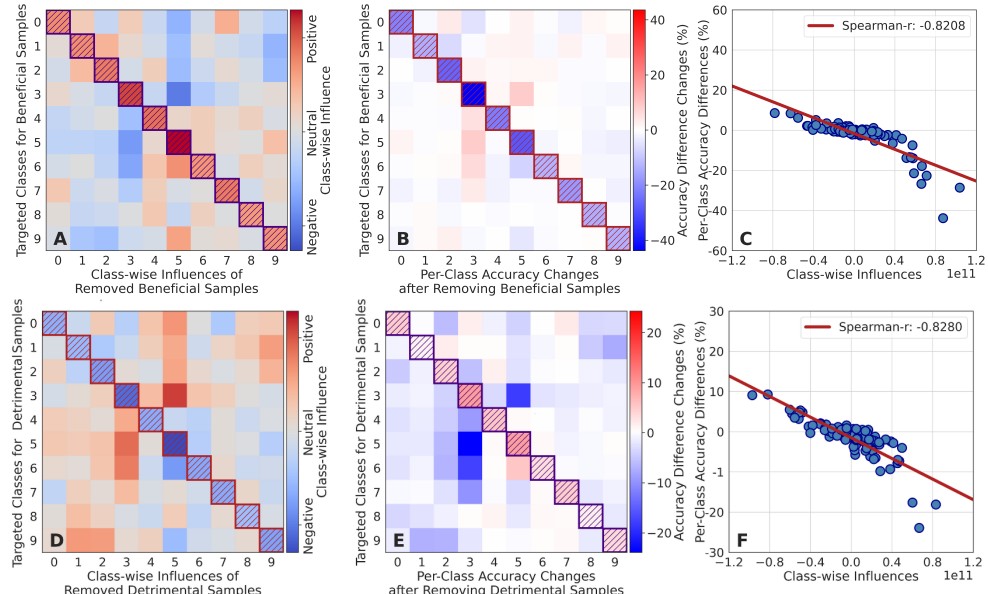

Figure 3: Real-world data experiments on *CIFAR10* (Krizhevsky et al., 2009). Subfigures **(A, D)** denote predicted category-wise influence and **(B, E)** actual accuracy changes when removing beneficial (top) or detrimental (bottom) samples. Subfigures **(C, F)** denote scatter plots showing the correlation between predicted influence and actual performance shifts.

category-wise influence distribution. As can be observed in subfigures **B** and **C** which showcase the category-wise influences and the samples corresponding to those influence values, respectively, this is indeed the case. Essentially, the mislabeled noisy samples from both categories possess negative influence for both categories, and hence, removing those should improve performance as expected. Additionally, a large number of non-noisy samples are positively/negatively influential for the blue/orange class (and vice-versa), validating our criterion further.

The bottom row consisting of subfigures **D-F** denotes a dataset that is non-linearly separable where performance improvements for both categories/classes cannot be jointly made. Here, the dataset consists of 350 samples for the blue category and 350 samples for the orange category, generated using a circular uniform distribution. For the orange samples, the radius of the distribution was changed depending on the angle from the center. In the ideal scenario, the Pareto frontier obtained via our category-wise influence functions should indicate that samples are positively influential for one class and negatively influential for the other class. Moreover, the samples with the maximum influence magnitude are those that appear around the decision boundary and ones that will inadvertently be misclassified due to the linear logistic regression classifier. In subfigures **E** and **F**, we can see that this holds true with the Pareto frontier visualized. The influence vectors of all training samples form a roughly straight line. As we demonstrate through these results, category-wise influence vectors can reveal the Pareto frontier for the two datasets accurately, and help users/developers make tradeoffs as required depending on the needs of their given application.

## 5    REAL-WORLD DATA EXPERIMENTS

In this section, we present our experimental results on real-world datasets in two parts: a validation of the effectiveness of category-wise influence functions and an evaluation of our PARETO-LP-GA for Pareto performance improvement.

### 5.1    CATEGORY-WISE INFLUENCE FUNCTIONS

We evaluate the category-wise influence using four widely adopted benchmark datasets with 4-10 categories: two vision datasets (*CIFAR-10* (Krizhevsky et al., 2009) and *STL-10* (Coates et al., 2011)) and two text datasets (*Emotion* (Saravia et al., 2018) and *AG_News* (Zhang et al., 2015)). For

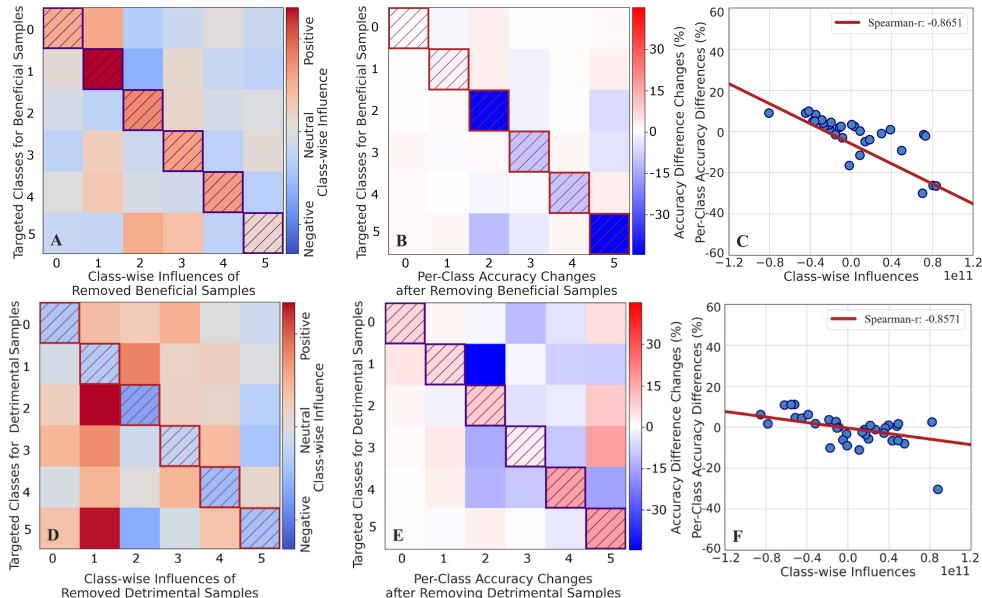

Figure 4: Real-world data experiments on *Emotion* (Saravia et al., 2018) text dataset.

calculating category-wise sample influence vectors, we employ EKFAC (Grosse et al., 2023) due to its fast implementation on deep models.

First, we evaluate whether category-wise influence serves as an effective indicator for measuring performance changes. To achieve this, we select the top 10% of beneficial and detrimental samples for each category, remove these samples from the training set, and retrain the model to observe the performance change. Figures 3 and 4 illustrate the performance changes for *CIFAR-10* and *Emotion*. Similar phenomena are observed for *STL-10* and *AG_News*, and these results are deferred to Appendix B.4. In these figures, Subfigures **A** and **D** illustrate the cumulative influence values of the removed training samples for beneficial and detrimental samples, respectively. Subfigures **B** and **E** display the corresponding performance changes after retraining without these samples. The results are presented as heat maps, with diagonal blocks representing the targeted categories, highlighted using bold boundaries and distinct textures.

The phenomena observed in the diagonal blocks reveal clear and consistent patterns: removing beneficial samples with positive influence on a target category leads to a performance drop in the corresponding category, while removing detrimental samples with negative influence results in a performance increase. This confirms the effectiveness of category-wise influence in predicting performance changes for both beneficial samples and detrimental samples. Beyond the diagonal blocks, the patterns are more mixed, as the selected samples based on one category may have varying impacts on other categories. To further analyze these non-diagonal patterns, we plot scatter diagrams of the cumulative influence for each category against their performance changes in Subfigures **C** and **F**. Subfigure **C** is derived from Subfigures **A** and **B**, while Subfigure **F** is based on Subfigures **D** and **E**. The Spearman correlation coefficient exceeding 0.8 indicates a strong relationship between sample influence and performance change, not only within the target category but also across categories. This demonstrates that category-wise influence functions can serve as an effective indicator for inferring performance changes. Moreover, they can be utilized to evaluate whether a classifier has reached its performance ceiling across different categories, providing valuable insights for targeted model improvement. We also present the numbers of beneficial and detrimental samples for each classes in Appendix B.4, where the beneficial samples mainly come from their category but detrimental samples come from other categories.

## 5.2 PERFORMANCE CEILING CHECK AND IMPROVEMENT

We evaluate our PARETO-LP-GA method using the *CIFAR10* dataset and ResNet model to showcase performance improvements for both *Direct Improvement* (*DI*) and *Course Correction* (*CC*) settings. The reason we chose *CIFAR10* for experiments was because during training multiple

Table 1: Comparison of category-wise accuracies for performance improvement in the *Direct Improvement* (left) and *Course Correction* (right) settings. Target classes are highlighted in blue. As can be observed, performance in target categories increases significantly while non-target classes see minimal reductions (or potential gains).

| Category | Epoch-10 | Epoch-11 (DI) | Change (%) | Category | Epoch-15 | Epoch-16 | Epoch-16 (CC) | Change (%) |
|---|---|---|---|---|---|---|---|---|
| 0 | 0.699 | 0.811 | **+16.02** | 0 | 0.876 | 0.876 | 0.889 | +1.48 |
| 1 | 0.888 | 0.881 | -0.78 | 1 | 0.866 | 0.868 | 0.870 | +0.23 |
| 2 | 0.667 | 0.743 | **+11.39** | 2 | 0.651 | 0.741 | 0.736 | -0.67 |
| 3 | 0.647 | 0.632 | -2.31 | 3 | 0.582 | 0.678 | 0.677 | -0.14 |
| 4 | 0.729 | 0.720 | -1.2 | 4 | 0.729 | 0.783 | 0.785 | +0.25 |
| 5 | 0.755 | 0.755 | +0.00 | 5 | 0.821 | 0.785 | 0.798 | **+1.65** |
| 6 | 0.802 | 0.845 | +5.73 | 6 | 0.859 | 0.859 | 0.855 | -0.46 |
| 7 | 0.849 | 0.848 | -0.11 | 7 | 0.885 | 0.837 | 0.848 | **+1.31** |
| 8 | 0.948 | 0.920 | -2.90 | 8 | 0.909 | 0.929 | 0.917 | -1.29 |
| 9 | 0.817 | 0.818 | +0.12 | 9 | 0.917 | 0.864 | 0.888 | **+2.77** |

epochs exhibit major Pareto tradeoffs across categories, while ensuring their is room for potential improvement. In contrast, for our text datasets (*Emotion* and *AG_News*), the NLP models achieved accuracies exceeding 90% across all classes within the first epoch, leading to little room for Pareto frontier improvement. Similarly, *STL-10* consists of *cleaner* images than *CIFAR10*, generally leading to better performance.

Before demonstrating performance improvements, we first examine whether the influence vectors of the training samples approximately lie on a hyperplane. To this end, we apply Principal Component Analysis (Wold et al., 1987) and compute the explained variance ratio of the first principal component. Across all targeted cases, this ratio consistently exceeds 0.2, indicating that the influence vectors do not fit a hyperplane and suggesting room for Pareto improvement.

We present results for *DI* and *CC* in Table 1. For *DI* in *CIFAR10*, we identified two categories (0 and 2) with relatively lower accuracy after observing performance at *Epoch 10* (0.699 for class-0, and 0.667 for class-2). These two classes will constitute our target categories. Then, we employ PARETO-LP-GA to obtain a weight set and apply weighted training to achieve *Epoch 11*. The results demonstrate that PARETO-LP-GA significantly enhances performance on the target categories, achieving improvements of 16.02% and 11.39% for classes 0 and 2, respectively. Notably, several non-target categories also experienced performance gains, such as category 6, which improved by 5.73%. However, for the non-target categories, performance degradation remains very minimal, showcasing the performance ceiling of the classifier via the Pareto improvement.

For *CC*, we identified *Epoch 16* as detrimental for classes 5, 7, and 9, where accuracies declined substantially (from *Epoch 15*). That is, between Epochs 15 and 16, performance for class 5 drops from $0.821 \rightarrow 0.785$; for class 7 drops from $0.885 \rightarrow 0.837$; and for class 9, drops from $0.917 \rightarrow 0.864$. Thus, these classes constitute our target categories for *CC*. We obtain the weight set using PARETO-LP-GA and after applying weighted training, we seek to obtain a new Epoch 16 that optimizes for the Pareto-optimal class tradeoffs as desired. This can be observed in our results – accuracy improves by 1.65%, 1.31%, and 2.77% in target categories 5, 7, and 9, respectively, reversing the original trends and bringing slight improvements. Additionally, class-0, class-1, and class-4 experienced 1.48%, 0.23%, and 0.25% performance enhancement, respectively, and there were only modest accuracy reductions in other categories ($\leq 1.29\%$). Overall, our proposed PARETO-LP-GA method demonstrates its ability to enhance training performance in specific target categories with minor accuracy tradeoffs in non-target categories.

## 6 CONCLUSION

In this paper, we extended the conventional influence function to a category-wise influence function and introduced the concept of an influence vector, which quantifies the impact of each training sample across all categories. Building on this formulation, we analyzed whether a classifier has already reached its maximum potential performance: formally, the Pareto frontier across categories, and design a linear programming–based sample reweighting framework for Pareto improvements. We validated the correctness of our performance-ceiling criterion through experiments on synthetic datasets, and further demonstrated the effectiveness of the proposed category-wise influence function on real-world datasets. Finally, we presented case studies that showcase how our sample reweighting approach can lead to tangible Pareto improvements across multiple categories.

# 7 REPRODUCIBILITY STATEMENT

We provide our code and implementation in a GitHub repository: `https://anonymous.4open.science/r/Classifier-Performance-Ceiling-C7FB/`. All the experiments were run multiple times to ensure reproducibility. Additionally, any parameters required for reproducibility (e.g., seeds) are provided in the repo and in Appendix A. The experiments were conducted on a Linux server with 8x NVIDIA A6000 GPUs.

# 8 ETHICS STATEMENT

In this paper, we study category-aware influence functions for classification models and propose methods that can utilize them for Pareto-optimal performance improvements across classes. Our work is beneficial for the community as it enables developers to navigate the performance ceiling of the classifier depending on the needs of their given application. Our methods are broadly applicable to any classification application domains (e.g., text and vision), further underscoring the positive impact of our methods.

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

APPENDIX

# A    DETAILED INFORMATION ON DATASETS AND MODEL TRAINING

We describe dataset details, model training, and other information used in the main paper, below.

## A.1    DATASETS

We introduce the synthetic datasets and real-world datasets used in the experiments.

### A.1.1    SYNTHETIC DATASETS

The datasets in Figure 2**A** is synthetic dataset that is linearly separable and contains noisy detrimental training samples. It consists of 300 blue class samples and 300 orange class samples, generated using a circular uniform distribution. Noise was added to the training set by choosing random points from each group, 50 from blue and 20 from orange, and then flipping their labels.

Similarly, the datasets in Figure 2**D** is is non-linearly separable where performance improvements for both categories cannot be joint made. This datasets consists of 350 samples for the blue category and 350 samples for the orange category, generated using a circular uniform distribution. For the orange samples, the radius of the distribution was changed depending on the angle from the center. There is no flip-labeled samples in this dataset.

### A.1.2    REAL-WORLD DATASETS

We use four widely adopted real-world benchmark datasets with 4-10 categories: two image datasets (*CIFAR-10* (Krizhevsky et al., 2009) and *STL-10* (Coates et al., 2011)) and two text datasets (*Emotion* (Saravia et al., 2018) and *AG_News* (Zhang et al., 2015)).

## A.2    MODELS

We train *bert-base-cased* (Devlin et al., 2018) for the *Emotion* and *AG_News* NLP datasets, and *ResNet-9* (He et al., 2016) for the *CIFAR10* and *STL-10* vision datasets in our experiments.

## A.3    PARAMETER DETAILS

Table 2 summarizes the hyperparameters used in all of our experiments.

Table 2: Hyperparameters for each dataset and algorithm.

| Dataset | Hyperparameters | Algorithm | Hyperparameters |
|---------|-----------------|-----------|-----------------|
| *CIFAR10* | train_batch_size: 512 | PARETO-LP-GA | train_batch_size: 512 |
|  | eval_batch_size: 1024 |  | eval_batch_size: 1024 |
|  | learning_rate: 0.4 |  | learning_rate: 0.0001 |
|  | weight_decay: 0.001 |  | weight_decay: 0.0001 |
|  | num_train_epochs: 25 |  | num_train_epochs: 20 |
| *STL-10* | train_batch_size: 512 |  | GA iterations: 20 |
|  | eval_batch_size: 1024 |  | Population_size: 24 |
|  | learning_rate: 0.4 |  | crossover_rate: 1.0 |
|  | weight_decay: 0.001 |  | mutation_rate: 0.25 |
|  | num_train_epochs: 30 |  | mutation_strength: 0.25 |
| *Emotion* | train_batch_size: 512 |  | num_elites: 6 |
|  | eval_batch_size: 1024 |  | num_mutated_elites: 6 |
|  | learning_rate: 0.4 |  | num_randoms: 6 |
|  | weight_decay: 0.001 |  | num_crossover_children: 6 |
|  | num_train_epochs: 25 |  |  |
| *AG_News* | train_batch_size: 512 |  |  |
|  | eval_batch_size: 1024 |  |  |
|  | learning_rate: 0.4 |  |  |
|  | weight_decay: 0.001 |  |  |
|  | num_train_epochs: 25 |  |  |

# B ADDITIONAL EXPERIMENTS

## B.1 ADDITIONAL EXPERIMENTS ON SYNTHETIC DATA

In the experiment shown in Figure 2, Subfigures **ABC**, the technique of identifying the noisy training points using their influence scores was discussed. Specifically, the removal of points which had negative influence scores with respect to both groups. In this experiment, we continue this path through to completion on the same dataset situation. We present the results in Figure 5. The dataset and model used are identical to the same from Subfigures **ABC** in Figure 2. The following method was applied: Create a linear regression model on the training set, calculate the category-aware influence scores of each training point, remove any training points with negative scores, and retrain the model on the new training set. It is shown that with each iteration of this technique, the model moves closer to the performance frontier.

## B.2 FINAL EPOCH REFINEMENT ON CIFAR10

We further applied our method at the final training stage (Epoch 19) of CIFAR-10 to demonstrate its effectiveness as a "final polish" for deployed models. Table 3 demonstrates the results. In the *Direct Improvement* setting, we targeted classes with lower relative performance (Classes 2, 3, 4, 5). We achieved significant gains, such as a **+4.88%** increase for Class 5 and **+3.71%** for Class 4. In the *Course Correction* setting, we targeted Classes 3 and 7, which experienced minor regression in the standard final epoch; our method successfully recovered their performance without retraining the model from scratch.

Table 3: Comparison of category-wise accuracies for **CIFAR-10 Final Epoch (Ep-19)** improvement in *Direct Improvement* (left) and *Course Correction* (right). Target classes are highlighted in blue.

| Class | Ep-18 | Ep-19 (DI) | Change (%) | Class | Ep-18 | Ep-19 | Ep-19 (CC) | Change (%) |
|-------|-------|------------|------------|-------|-------|-------|------------|------------|
| 0 | 0.904 | 0.895 | -1.00 | 0 | 0.904 | 0.897 | 0.897 | +0.00 |
| 1 | 0.838 | 0.868 | +3.50 | 1 | 0.838 | 0.871 | 0.863 | -0.92 |
| 2 | 0.703 | 0.704 | **+0.14** | 2 | 0.703 | 0.725 | 0.711 | -1.93 |
| 3 | 0.724 | 0.733 | **+1.24** | 3 | 0.724 | 0.700 | 0.703 | **+0.42** |
| 4 | 0.781 | 0.810 | **+3.71** | 4 | 0.781 | 0.786 | 0.784 | -0.25 |
| 5 | 0.758 | 0.795 | **+4.88** | 5 | 0.758 | 0.810 | 0.814 | +0.49 |
| 6 | 0.869 | 0.879 | +1.15 | 6 | 0.869 | 0.872 | 0.879 | +0.80 |
| 7 | 0.832 | 0.819 | -1.56 | 7 | 0.832 | 0.827 | 0.831 | **+0.48** |
| 8 | 0.918 | 0.919 | +0.11 | 8 | 0.918 | 0.923 | 0.920 | -0.32 |
| 9 | 0.881 | 0.898 | +1.92 | 9 | 0.881 | 0.892 | 0.905 | +1.56 |

## B.3 EXPERIMENTS ON IMAGENET SUBSET

To validate the scalability of our approach, we conducted experiments on a subset of ImageNet (55k training samples, 11 Superclasses (Tsipras et al., 2020)). We illustrate two scenarios in Table 4: *Direct Improvement* (Left) targeting Superclasses 6, 8, 9, and 10; and *Course Correction* (Right) targeting Superclasses 0, 2, 3, and 8 to recover performance lost during standard training.

In the Direct Improvement setting, we observe a gain for Class 10 (**+13.7**) and Class 7 (**+8.4**), boosting the overall average accuracy by **2.0**. In the Course Correction setting, our method successfully reversed the performance degradation observed in standard Epoch 19, improving Class 0 by **13.1** relative to the baseline. We demonstrate the performance gains in a bar plot in Figure 6.

## B.4 ADDITIONAL EXPERIMENTS ON REAL-WORLD DATA

Figures 7 and 8 illustrate the performance changes for *STL-10* and *AG_News* by removing beneficial and detrimental samples according to influence vectors. They provide similar phenomena with the results on *CIFAR-10* and *Emotion*, which we illustrate in the main paper.

Figure 9 present the numbers of beneficial and detrimental samples for each classes on *CIFAR-10* dataset, where the beneficial samples mainly come from their category but detrimental samples come from other categories.

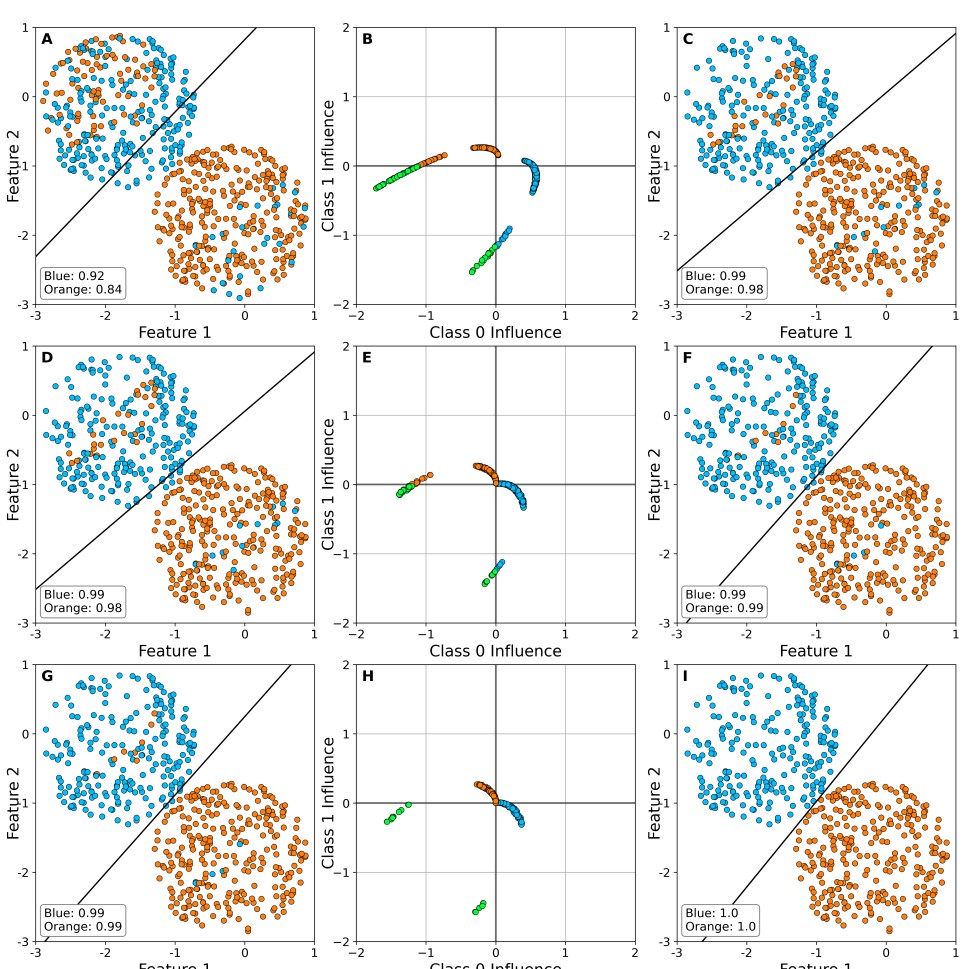

Figure 5: Experiment demonstrating the use of the category-wise influence function for dataset augmentation. The dataset used in this demonstration is identical to that in the top row of Figure 1. Each row displays the state of the dataset, the changes that our method will make, and the result, of the dataset trimming procedure discussed in Appendix A. Subfigures A, D, and G display the state of the training dataset. The color of each point indicates its label. The linear decision boundary is drawn, and its accuracy across both classes is shown in the legend. Subfigures B, E, and H show the category-wise influence score of each training point. Training data points in green are indicated by the score to be detrimental to model performance on both classes. These points will be removed by the improvement procedure. Subfigures C, F, and H show the training dataset after removal. The resulting decision boundary and its accuracy is also indicated. Note how the noise furthest from the decision boundary is removed first, since these have the largest effect on the decision boundary. Additionally, through each iteration, the accuracy of the linear model is improved when using the trimmed dataset. After three iterations, all noise has been removed and the model is at the performance ceiling.

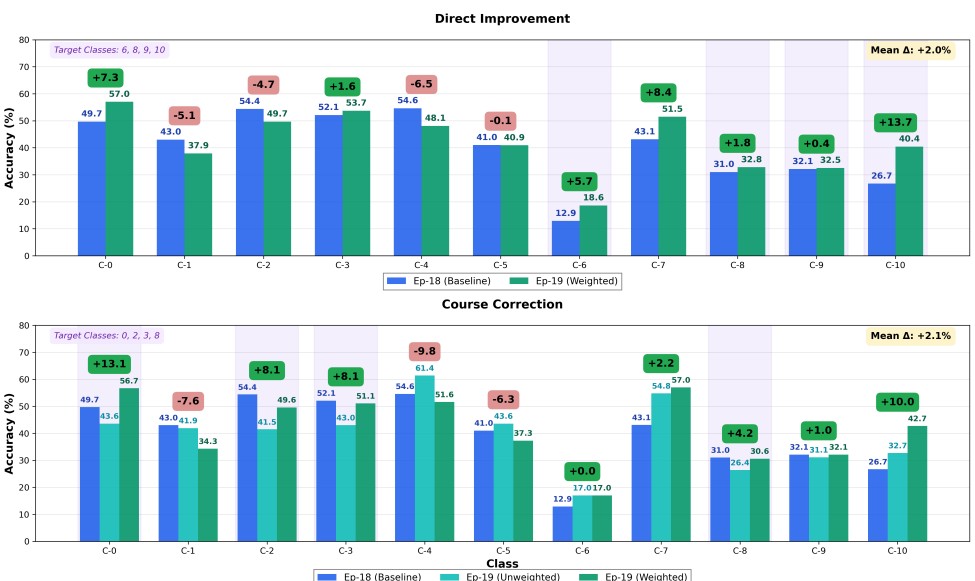

Figure 6: ImageNet performance shifts.

Table 4: Comparison of category-wise accuracies for the **ImageNet Subset** in *Direct Improvement* (left) and *Course Correction* (right). Target classes are highlighted in blue.

| Class | Ep-18 | Ep-19 (DI) | Change | Class | Ep-18 | Ep-19 | Ep-19 (CC) | Change |
|-------|-------|-----------|--------|-------|-------|-------|-----------|--------|
| 0 | 49.7 | 57.0 | +7.3 | 0 | 49.7 | 43.6 | 56.7 | **+13.1** |
| 1 | 43.0 | 37.9 | -5.1 | 1 | 43.0 | 41.9 | 34.3 | -7.6 |
| 2 | 54.4 | 49.7 | -4.7 | 2 | 54.4 | 41.5 | 49.6 | **+8.1** |
| 3 | 52.1 | 53.7 | +1.6 | 3 | 52.1 | 43.0 | 51.1 | **+8.1** |
| 4 | 54.6 | 48.1 | -6.5 | 4 | 54.6 | 61.4 | 51.6 | -9.8 |
| 5 | 41.0 | 40.9 | -0.1 | 5 | 41.0 | 43.6 | 37.3 | -6.3 |
| 6 | 12.9 | 18.6 | **+5.7** | 6 | 12.9 | 17.0 | 17.0 | +0.0 |
| 7 | 43.1 | 51.5 | +8.4 | 7 | 43.1 | 54.8 | 57.0 | +2.2 |
| 8 | 31.0 | 32.8 | **+1.8** | 8 | 31.0 | 26.4 | 30.6 | **+4.2** |
| 9 | 32.1 | 32.5 | **+0.4** | 9 | 32.1 | 31.1 | 32.1 | +1.0 |
| 10 | 26.7 | 40.4 | **+13.7** | 10 | 26.7 | 32.7 | 42.7 | +10.0 |

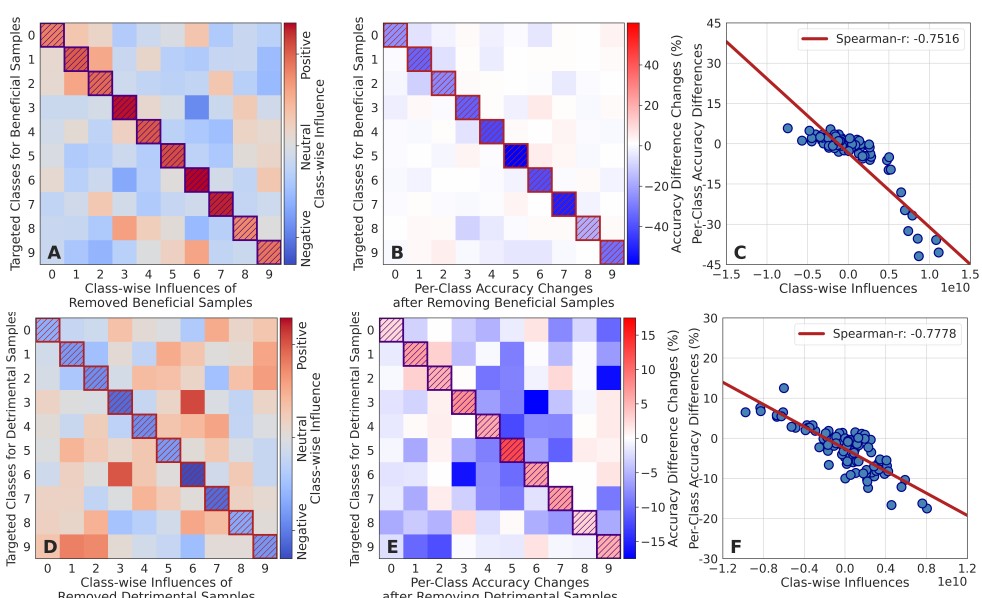

Figure 7: Real-world data experiments on *STL-10* (Coates et al., 2011) image dataset.

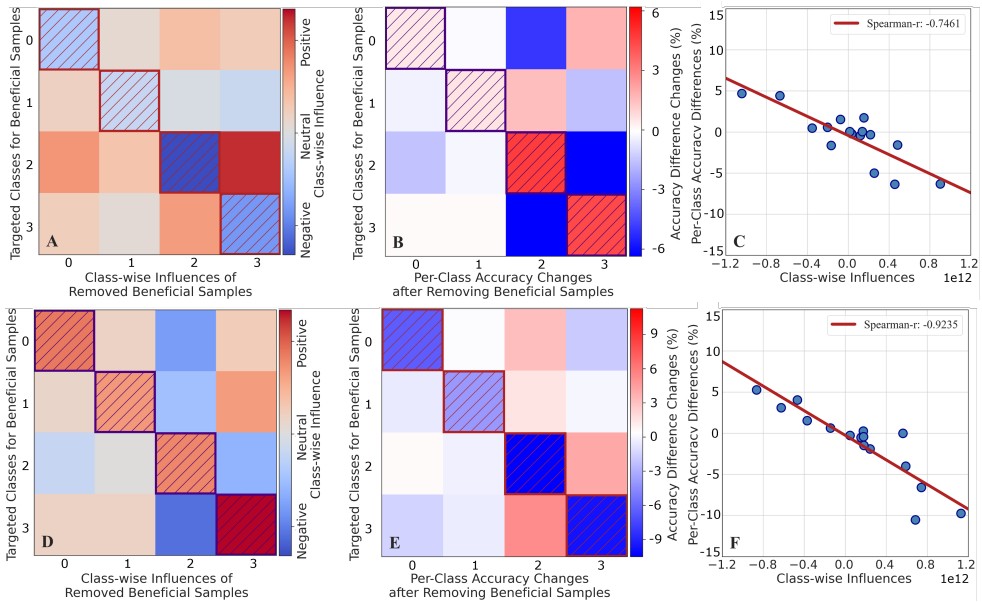

Figure 8: Real-world data experiments on *AG_News* (Zhang et al., 2015) text dataset.

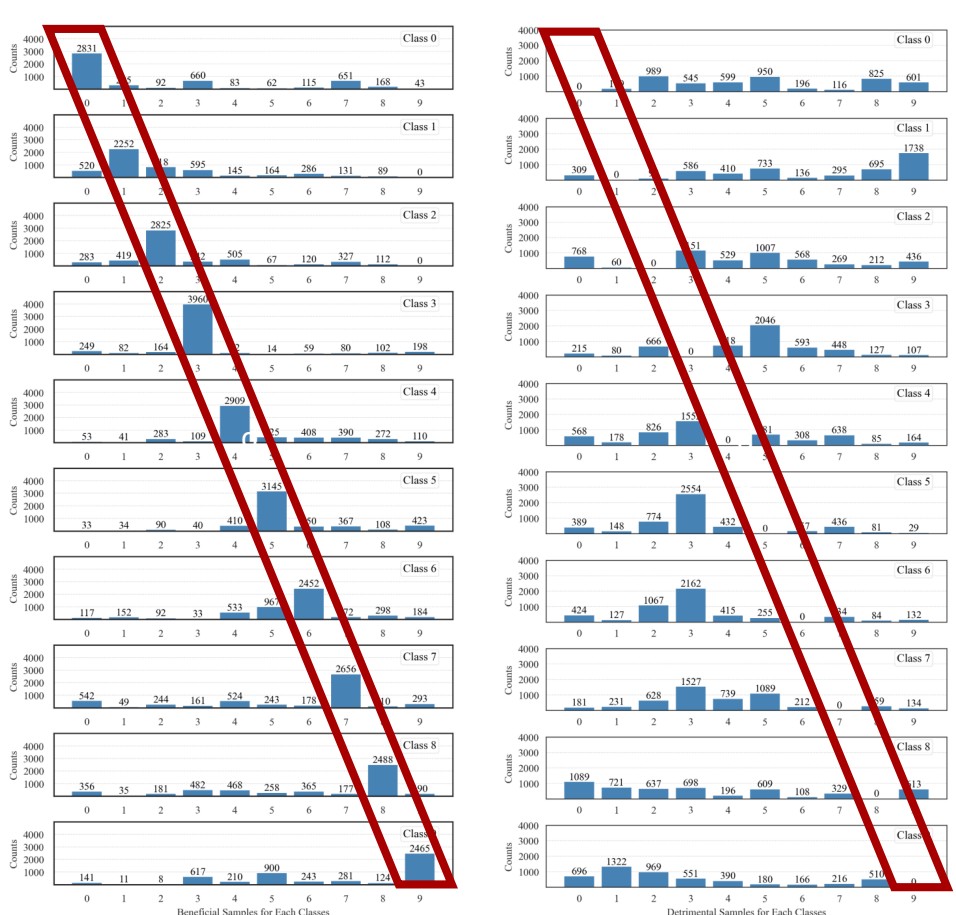

Figure 9: The numbers of beneficial and detrimental samples for each classes on *CI-FAR10* (Krizhevsky et al., 2009) image dataset.

