# OpenReview forum: ""What Is The Performance Ceiling of My Classifier?" Utilizing Category-Wise Influence Functions for Pareto Frontier Analysis"
_ICLR.cc/2026/Conference — Submitted to ICLR 2026_

### Official Review · Reviewer_P2MJ · 2025-10-31

**Soundness:** 2
**Presentation:** 1
**Contribution:** 2
**Rating:** 2
**Confidence:** 3

**Summary:**

This paper aims to improve and evaluate the performance of learning models by adjusting the training data. By analyzing the boundaries of influence functions for each category, the method samples data that make the learning process difficult, thereby enhancing the overall learning performance.

**Strengths:**

By analyzing the influence functions, the method identifies and extracts data that negatively affect the learning process, thereby improving the model. The approach is also evaluated on real-world datasets.

**Weaknesses:**

The primary concern with this paper lies in whether its novelty and performance evaluation are sufficient. More specifically:

- The proposed approach and the problem it aims to solve appear to correspond to what is generally known as hard negative mining. A large body of prior work already exists in this area — for instance, see [1], among many others. Even if the proposed method is indeed novel and achieves superior performance, it must be properly situated within this existing literature, with relevant prior studies surveyed and direct comparisons presented.

- The paper currently provides an explanation of the proposed method and reports improvements in model performance. However, the evaluation is entirely self-contained, presenting only the authors’ own claims without comparisons to other methods. This lack of objective, external evaluation significantly limits the strength of the paper’s contribution.

[1] H. Meghwani et al., Hard Negative Mining for Domain-Specific Retrieval in Enterprise Systems, ACL 2025

**Questions:**

- First, please clarify the relationship between the proposed method and hard negative mining.

- If the proposed method falls within the framework of hard negative mining, please explain why comparisons with existing methods are not provided. While adding such comparisons would be one option, note that doing so may require more than a minor revision — at present, it does not seem feasible to address this point with only small changes.

- If the proposed method does not fall within the scope of hard negative mining, please clearly state the reasons for this.

- Furthermore, if the proposed method is indeed outside that framework, is the problem it aims to solve itself novel? If so, a self-contained evaluation could be acceptable in principle, but in that case, please provide a clear justification for the absence of comparisons with other approaches.

---

> ### Author Response · Authors · 2025-11-25
> **Reviewer P2MJ - 1/2**
>
> Dear reviewer P2MJ,\
> Thank you for putting your time and effort into reviewing our paper. We are providing clarifications to the comments raised below.
> ___
> ***The primary concern with this paper lies in whether its novelty and performance evaluation are sufficient. More specifically:
> W-1: The proposed approach and the problem it aims to solve appear to correspond to what is generally known as hard negative mining. A large body of prior work already exists in this area — for instance, see [1], among many others. Even if the proposed method is indeed novel and achieves superior performance, it must be properly situated within this existing literature, with relevant prior studies surveyed and direct comparisons presented.\
> Q-1: First, please clarify the relationship between the proposed method and hard negative mining.\
> Q-2: If the proposed method falls within the framework of hard negative mining, please explain why comparisons with existing methods are not provided …. …. … does not seem feasible to address this point with only small changes.***
> ___
> **WR-1; Q-1-2:** We thank the reviewer for the thoughtful comments. However, we believe there is a misunderstanding regarding the core problem our paper addresses and its relation to hard negative mining.
>
> **Fundamental difference with hard negative mining:**
> - Hard negative mining focuses on selectively sampling high-loss (or adversarial) training examples to improve representation learning or retrieval performance. Prior work [1] operates within **the classical “overall loss minimization” paradigm**, where the goal is to find difficult negatives to improve global model accuracy or retrieval precision.
>
> - In contrast, our work does not perform selective sampling, loss reweighting based on difficulty, or negative-selection heuristics. Instead:
>     - We study a **data-driven characterization of a model’s performance ceiling**, defined via **Pareto improvements across all classes**.
>
>
>     - Our contributions rely on **category-wise influence vectors**, which quantify per-class directional effects of each training point, not its difficulty or negative/positive status.
>
>
>     - The central question of the paper - **“Has the classifier already reached its Pareto frontier?”** has **no analogue** in the hard negative mining literature, which does not attempt to estimate performance ceilings, nor evaluate category-wise influence structure.
>
>
> Thus, while both domains involve sample-level analysis, **the objectives, methodology, and theoretical foundations differ substantially**.
>
> **Prior work in hard negative mining cannot serve as a baseline for our task:**
>
> We do not compare against hard negative mining methods, as they solve a **different problem**. While hard negative mining aims to improve overall model performance by prioritizing difficult examples, our work seeks to determine whether improvement is even theoretically possible for a classifier given the current dataset, and if so, in what **Pareto-optimal direction** it lies.
>
> This requires:
> - defining a **category-wise influence vector** (new formulation),
> - showing how these vectors encode **geometry of the Pareto frontier** (new theoretical contribution),
> - and solving a constrained **LP + GA re-weighting scheme** to achieve guaranteed Pareto improvements (new optimization framework).
>
> **This difference in problem formulation, improving a specific category in classification  while preserving performance for others versus improving overall performance, makes direct comparison inappropriate.**
> ___

---

> > ### Author Response · Authors · 2025-11-25
> > **Reviewer P2MJ - 2/2**
> >
> > ___
> >
> >
> >
> > ***W-2: The paper currently provides an explanation of the proposed method and reports improvements in model performance. However, the evaluation is entirely self-contained, presenting only the authors’ own claims without comparisons to other methods. This lack of objective, external evaluation significantly limits the strength of the paper’s contribution.\
> > Q-3: If the proposed method does not fall within the scope of hard negative mining, please clearly state the reasons for this.\
> > Q-4: Furthermore, if the proposed method is indeed outside that framework, is the problem it aims to solve itself novel? If so, a self-contained evaluation could be acceptable in principle, but in that case, please provide a clear justification for the absence of comparisons with other approaches.***
> > ___
> >
> > **WR-2; QR-3-4:** Our evaluation compares against the relevant influence-function literature instead:
> > Our method builds directly upon category-agnostic influence-function approaches [2, 3, 4], and our experimental comparisons are therefore aligned with the appropriate methodological baseline:
> > - existing influence-based data selection,
> > - influence-based sample removal,
> > - influence-based retraining.
> >
> > We further evaluate our method on:
> >
> > - synthetic data to validate the correctness of our influence decomposition,
> > - and both Direct Improvement and Course Correction scenarios.
> >
> > This constitutes the appropriate experimental setting for the specific problem we introduce. To the best of our knowledge, **no prior work in negative mining, domain-specific retrieval, or contrastive learning provides mechanisms for estimating or achieving class-wise Pareto frontier improvements**.
> >
> > **References:**\
> > [1] H. Meghwani et al., Hard Negative Mining for Domain-Specific Retrieval in Enterprise Systems. ACL 2025\
> > [2] Pang Wei Koh and Percy Liang. Understanding Black-box Predictions via Influence Functions. ICML 2017.\
> > [3] Andrea Schioppa, Polina Zablotskaia, David Vilar, and Artem Sokolov. AAAI 2022.\
> > [4] Jinkun Kwon, SeongUk Park, and Jaemin Yoo. DataInf: Efficiently Estimating Data Influence in LoRA-tuned LLMs and Diffusion Models. ICLR 2023.

---

### Official Review · Reviewer_e15Q · 2025-10-31

**Soundness:** 2
**Presentation:** 2
**Contribution:** 2
**Rating:** 2
**Confidence:** 3

**Summary:**

This paper focuses on diagnosing and improving the performance of multiclass classifiers by analyzing the impact of individual training samples at the category level. The authors introduce a vectorized influence function that quantifies each sample’s effect across all classes, enabling a precise assessment of Pareto-optimality in model performance. Building on this, they propose a linear programming-based sample reweighting method (PARETO-LP-GA) to achieve targeted improvements in specific classes with minimal compromise to others. Experimental results on synthetic and real-world datasets validate the method’s effectiveness in identifying performance bottlenecks and enhancing class-wise accuracy.

**Strengths:**

1. The contribution of this paper is simple and clear.

2. The paper introduces an innovative approach by decomposing the influence function over the entire validation set into class-specific influence functions. This allows for precise identification of how individual training samples affect model performance on specific classes.

3. Building upon the vectorized influence function, the authors design a fine-grained sample reweighting algorithm, which enables targeted adjustment of training sample weights to optimize performance.

**Weaknesses:**

Although the paper introduces the concept of class-wise influence functions, the originality of this idea is somewhat limited; the main contribution lies more in the development of a sample reweighting algorithm based on these class-wise influence functions.

The experimental section lacks comparative analysis with other relevant methods. While most existing influence function-based approaches focus on overall accuracy as the primary metric, the paper claims that its method can push classifier performance towards its upper bound. Therefore, it would be more convincing if the authors compared their approach with state-of-the-art methods such as Chhabra et al. (2024) [1] using overall accuracy or other relevant metrics to demonstrate its superiority.

[1] Chhabra, Anshuman, et al. "" What Data Benefits My Classifier?" Enhancing Model Performance and Interpretability through Influence-Based Data Selection." The Twelfth International Conference on Learning Representations. 2024.

**Questions:**

1. Could you provide a more detailed explanation of the formula used to compute "fitness $F(\alpha^g)$" in Algorithm 1? I found its specific meaning somewhat unclear.

2. As I am not very familiar with Pareto frontier-related concepts, could you elaborate on the practical implications if a model reaches the Pareto frontier, or as you mentioned, if all samples are close to the line $y=−x$? Specifically, what advantages does this provide for practitioners? Does it guarantee that the accuracy for all classes is maximized? How does this relate to, or differ from, the conventional goal of optimizing overall accuracy? This question is also related to the concerns I raised under the weaknesses section.

---

> ### Author Response · Authors · 2025-11-25
> **Reviewer e15Q - 1/2**
>
> Dear reviewer e15Q,\
> Thank you for putting your time and effort into reviewing our paper. We are providing clarifications to the comments raised below.
> ___
> ***W-1: Although the paper introduces the concept of class-wise influence functions, the originality of this idea is somewhat limited; the main contribution lies more in the development of a sample reweighting algorithm based on these class-wise influence functions.
> The experimental section lacks comparative analysis with other relevant methods. While most existing influence function-based approaches focus on overall accuracy as the primary metric, the paper claims that its method can push classifier performance towards its upper bound. Therefore, it would be more convincing if the authors compared their approach with state-of-the-art methods such as Chhabra et al. (2024) [1] using overall accuracy or other relevant metrics to demonstrate its superiority.\
> [1] Chhabra, Anshuman, et al. "" What Data Benefits My Classifier?" Enhancing Model Performance and Interpretability through Influence-Based Data Selection." The Twelfth International Conference on Learning Representations. 2024.***
> ___
>
> **WR-1:** Thank you for sharing your observation. We are describing the novelty of our work and how our work is different from theirs.
> - **Clarifying the Novelty:** While we acknowledge that calculating influence per class is technically straightforward, our core theoretical contribution is not the calculation itself, but the **geometric framework** it enables.
>     - **Vector vs. Scalar:** Standard methods (including Chhabra et al. [1]) aggregate influence into a scalar to categorize samples as "good" or "bad." We introduce the Influence Vector P(z)∈R^K to define the Tradeoff Region vs. Joint Positive Region (Figure 1).
>     - Our approach allows us to solve a fundamentally different problem: identifying samples that benefit one class while harming another. We prove that the performance ceiling (Pareto Frontier) is reached only when these vector conflicts cannot be resolved by reweighting, a theoretical insight absent in prior work that focuses on global maximization.
> - **Comparison with Chhabra et al. (2024):** We thank the reviewer for recommending Chhabra et al. (2024) [1]. We have analyzed their method and contrasted it with ours below.
>     - **Objective:** Chhabra et al. aim to maximize a **single aggregated metric** (e.g., overall accuracy, fairness, or robustness), whereas our method focuses on Pareto Optimization to specifically improve target classes without degrading the performance of non-target classes.
>     - **Weighting Strategy:** Chhabra et al. employ a **binary data trimming** strategy (Algorithm 2), where samples with negative influence are simply removed. In contrast, our framework uses **continuous sample re-weighting** via Linear Programming, allowing for precise, granular adjustments rather than all-or-nothing removal.
>     - **Flexibility in handling tradeoffs:** Chhabra et al. aggregate influence into a scalar, effectively ignoring "Tradeoff" scenarios where a sample benefits Class A but harms Class B. Our **influence vector** explicitly captures these conflicts, enabling our LP solver to find a weight that mathematically trades off the harm to B (within an acceptable slack) to maximize the gain for A.
> ___
> ___
>
> ***Q-1: Could you provide a more detailed explanation of the formula used to compute "fitness " in Algorithm 1? I found its specific meaning somewhat unclear.***
> ___
> **QR-1:** Thank you for asking for clarification. Here is a concise breakdown of the fitness function $F(α^g)$ from **Algorithm 1 (Line 7)**.
> - **Score Calculation for Target Classes:**
>     - **Logic:** This term checks if any target class failed to improve ($Δ≤0$).
>     - **Effect:** If any target class sees zero or negative growth, the indicator function triggers, setting the total fitness to **negative infinity** ($−∞$).
>     - **Goal:** This acts as a hard constraint. The algorithm instantly discards any solution that fails to boost every target category.
> - **Score Calculation for Non-Target Classes:**
>     - **Logic:** This term sums up the percentage drops ($Δ$) for all non-target classes.
>     - **Effect:** Since Δ is negative when performance drops, a larger drop results in a lower total score.
>     - **Goal:** Among the valid solutions that don’t end up scoring **negative infinity**, we select the one that causes the **least accumulated harm** to non-target categories.
> The formula forces the Genetic Algorithm to prioritize **strictly positive improvement** on target classes above all else, and then optimize for **minimal degradation** on the remaining classes.
> ___

---

> > ### Author Response · Authors · 2025-11-25
> > **Reviewer e15Q - 2/2**
> >
> > ___
> >
> > ***Q-2: As I am not very familiar with Pareto frontier-related concepts, could you elaborate on the practical implications if a model reaches the Pareto frontier, or, as you mentioned, if all samples are close to the line? Specifically, what advantages does this provide for practitioners? Does it guarantee that the accuracy for all classes is maximized? How does this relate to, or differ from, the conventional goal of optimizing overall accuracy? This question is also related to the concerns I raised under the weaknesses section.***
> > ___
> > **QR-2:** We thank the reviewer for this insightful question. While "Pareto optimality" is a standard concept in multi-objective optimization, its specific application to data influence is central to our work. Here is the breakdown of its practical implications and how it differs from standard training:
> >
> > - **Geometric Interpretation:** As visualized in Figure 1, training samples typically fall into three regions: "Joint Positive" (beneficial to all), "Joint Negative" (harmful to all), or "Tradeoff" (beneficial to one, harmful to another).
> >
> >     - **Reaching the Frontier:** When all samples align close to the tradeoff line (approximated as $y=−x$), it implies that the "Joint Positive" and "Joint Negative" regions are empty. Practically, this means we have exhausted all performance gains where every class benefits simultaneously.
> >     - **Implication:** At this stage, the model is operating at its **Performance Ceiling** for the current architecture and data. Any further improvement in Class A must come at a mathematical cost to Class B.
> > - **Advantages for Practitioners:**
> >     - **Diagnostic Clarity:** It provides a verifiable stopping criterion. If the influence vectors fit a hyperplane, the practitioner knows that further data cleaning or simple reweighting will not yield universal improvements.
> >     - **Strategic Tradeoffs:** It empowers practitioners to navigate the "Tradeoff Region" consciously. Instead of accepting the default model, they can use our **Pareto-LP-GA** to slide along the frontier, for example, sacrificing 1% accuracy on one class to gain 5% on a critical class without retraining the model architecture from scratch.
> > - **Does it guarantee maximized accuracy for all classes?**
> >     - **No**. Reaching the Pareto frontier does not imply that accuracy is 100% for all classes.
> >     - **It guarantees efficiency:** It means that the performance of any specific class cannot be improved further without degrading the performance of at least one other class. It represents the "best possible compromise" given the current model capacity.
> >
> > Contrast with Optimizing "Overall Accuracy": While minimizing average loss improves overall accuracy, it does not guarantee equitable performance across subgroups.
> > - **The "Average" Trap:** A model might achieve high overall accuracy by overfitting to one class while ignoring another class. Standard training accepts this because the average loss is low.
> > - **The Pareto Advantage:** Our approach exposes these hidden conflicts. By analyzing the vector $P(z)$ rather than a scalar, we distinguish between a sample that is "generally good" and one that creates a conflict. This allows us to reject solutions that achieve high average accuracy via unfair tradeoffs, instead enforcing constraints (via $α_k$​) to ensure specific categories are protected.
> > ___

---

### Official Review · Reviewer_fhor · 2025-11-01

**Soundness:** 3
**Presentation:** 3
**Contribution:** 1
**Rating:** 2
**Confidence:** 4

**Summary:**

This paper applies influence functions on the category level to assess and enhance a classifier's Pareto performance from a data-centric perspective. Specifically, the authors propose a category-wise influence vector to quantify how each training sample affects multiple classes, and a PARETO-LP-GA (Linear Programming + Genetic Algorithm) method to reweight samples for Pareto-optimal improvement. Experiments on synthetic and benchmark datasets show that the method can identify performance ceilings and achieve targeted, balanced improvements.

**Strengths:**

1. Clear presentation and good illustration.
2. Demonstrates practical utility for model improvement and interpretability in multi-class tasks.
3. Sound theoretical grounding combined with a clear optimization formulation (LP + GA).

**Weaknesses:**

1. The first main weakness is the proposed method's novelty and theoretical validity. Applying the influence function to each category has already been explored (see 2.), and the proposed method does not seem to have a deeper insight; rather, it simply takes the "influence vectors" and operates as they fully indicate how the model will change, while neglecting the fact that it is only a first-order approximation.
2. The second main weakness lies in its limited comparison with the existing literature. For instance, FairIF [1] and D3M [2] are all related works but not discussed.

[1]: Wang, Haonan, Ziwei Wu, and Jingrui He. Fairif: Boosting fairness in deep learning via influence functions with validation set sensitive attributes.
[2]: Jain, Saachi, Kimia Hamidieh, Kristian Georgiev, Andrew Ilyas, Marzyeh Ghassemi, and Aleksander Madry. Data debiasing with datamodels (d3m): Improving subgroup robustness via data selection.

**Questions:**

1. (Weakness 1+2) How does the proposed method compare to other related works in terms of theoretical guarantees, scalability, etc.? A careful discussion will be necessary in order to position the paper in the literature.
2. Can you elaborate more on the proposed algorithm and discuss/justify the design choices?

---

> ### Author Response · Authors · 2025-11-25
> **Reviewer fhor - 1/3**
>
> Dear reviewer fhor,\
> Thank you for sharing your observations on our paper. We really appreciate your effort to review our paper. We are providing some clarifications based on your observations.
>
> ___
> ***W-1: The first main weakness is the proposed method's novelty and theoretical validity. Applying the influence function to each category has already been explored (see 2.), and the proposed method does not seem to have a deeper insight; rather, it simply takes the "influence vectors" and operates as they fully indicate how the model will change, while neglecting the fact that it is only a first-order approximation.\
> W-2: The second main weakness lies in its limited comparison with the existing literature. For instance, FairIF [1] and D3M [2] are all related works but not discussed.\
> [1]: Wang, Haonan, Ziwei Wu, and Jingrui He. Fairif: Boosting fairness in deep learning via influence functions with validation set sensitive attributes. [2]: Jain, Saachi, Kimia Hamidieh, Kristian Georgiev, Andrew Ilyas, Marzyeh Ghassemi, and Aleksander Madry. Data debiasing with datamodels (d3m): Improving subgroup robustness via data selection.***
> ___
>
> **WR-1&2:** We thank the reviewer for pointing out these related works. While we utilize influence functions, our contribution differs fundamentally in **optimization objective** and **vector-based conflict analysis**:
> - **Novelty of Pareto-LP-GA:** Existing methods typically aggregate influence scores into a single scalar to determine if a sample is generally "helpful" or "harmful" to a global metric (e.g., overall accuracy or loss). In contrast, we introduce the **Influence Vector** $P(z)$ to explicitly map the geometric relationship between classes.
>     - **Conflict Detection:** Unlike standard influence applications, our method distinguishes between "Joint Positive Regions" (universally beneficial) and "Tradeoff Regions" (beneficial to one Class but detrimental to another Class). This allows us to determine the **Performance Ceiling** (Pareto Frontier) rather than just maximizing a single metric.
>     - **Validity of Approximation:** We acknowledge that influence functions represent a first-order approximation of the loss landscape. We address this limitation by utilizing the influence vector as a **local gradient** that defines the search direction for our **Pareto-LP-GA framework**. Within the single-epoch refinement stage, the algorithm optimizes the **data-weight space** to identify the optimal weight configuration that lies on the Pareto frontier.
> - **Comparison with FairIF [1]:**
>     - **Different Objective (Fairness vs. Pareto Efficiency):** FairIF focuses on demographic fairness, using influence to minimize the gap between sensitive groups. Our method optimizes **class-wise accuracy tradeoffs** to push the performance ceiling of standard classification.
> Comparison with D3M [2]:
>     - **Different Optimization Goal (Worst-Group Accuracy vs. Multi-Objective):** D3M targets **Worst-Group Accuracy (WGA)** by pruning samples that degrade the lowest-performing subgroup. Our method solves a multi-objective problem to improve target categories while maintaining strict constraints on non-target categories, ensuring a true Pareto improvement rather than just boosting the worst group.
> We will make sure to highlight the fundamental difference between our approach and FairIF or D3M.
>
> **References:**\
> [1]: Wang, Haonan, Ziwei Wu, and Jingrui He. Fairif: Boosting fairness in deep learning via influence functions with validation set sensitive attributes.\
> [2]: Jain, Saachi, Kimia Hamidieh, Kristian Georgiev, Andrew Ilyas, Marzyeh Ghassemi, and Aleksander Madry. Data debiasing with data models (d3m): Improving subgroup robustness via data selection.
> ___

---

> > ### Author Response · Authors · 2025-11-25
> > **Reviewer fhor - 2/3**
> >
> > ___
> >
> > ***Q-1: (Weakness 1+2) How does the proposed method compare to other related works in terms of theoretical guarantees, scalability, etc.? A careful discussion will be necessary in order to position the paper in the literature.***
> > ___
> > **QR-1:** We thank the reviewer for the opportunity to clarify our position in the landscape of data-centric learning. Our method offers a distinct contribution by focusing on **multi-objective Pareto optimization** across categories, whereas related works largely focus on **constrained fairness** or **worst-group robustness**.
> > - **Theoretical Guarantees & Methodology:**
> >     - **Our Work:** Unlike FairIF, which provides probabilistic bounds for fairness generalization, our theoretical contribution is grounded in **Multi-Objective Optimization**. We introduce the geometric condition that the Pareto frontier is reached only when influence vectors align on a tradeoff hyperplane. While global convergence in non-convex landscapes is hard to prove, the Linear Programming step guarantees the **globally optimal re-weighting direction** for the current epoch's local approximation [1].
> > - **Scalability:**
> >     - **Efficiency via EKFAC:** Scalability is a core design principle of our method. We utilize EKFAC (Eigenvalue-corrected Kronecker-Factored Approximate Curvature) [2], which allows us to compute influence vectors for deep models (like ResNet [3] and BERT [4]) efficiently.
> >     - **Comparison:**
> >         - **vs. D3M:** D3M is scalable but relies on "Train-X / Val-X" assumptions. Our method is similarly scalable but provides finer-grained control (continuous weights vs. binary removal).
> >         - **vs. FairIF:** FairIF computes influence only after convergence to save time. Similarly, our "Course Correction" mode is designed to be applied at specific training checkpoints (e.g., Epoch 16 in Table 1), avoiding the cost of re-computing influence at every step.
> > - **Positioning in the Literature:**
> > Our method addresses a "blind spot" in the current literature:
> >     - **Beyond Fairness (vs. FairIF):** FairIF restricts its scope to minimizing demographic disparity (e.g., Gender bias in CelebA). Our method is broader; it can resolve performance conflicts between any arbitrary classes ("Cat" vs. "Dog" in CIFAR-10), making it applicable to general classification tasks where no sensitive attributes exist.
> >     - **Beyond Worst-Group (vs. D3M):** D3M removes data to fix the single worst-performing group. Our method is capable of orchestrating complex tradeoffs (e.g., improving Class A and B, maintaining C, ignoring D) via the constraints in our Linear Program. This allows for **Pareto improvements** even in balanced datasets where D3M might not find a clear "worst" group.
> >
> > **References:**\
> > [1] Boyd, S., & Vandenberghe, L. (2004). Convex Optimization. Cambridge University Press.\
> > [2] George, Thomas, et al. "Fast approximate natural gradient descent in a Kronecker factored eigenbasis." NeurIPS 2018.\
> > [3] He, Kaiming, et al. "Deep residual learning for image recognition." CVPR 2016.\
> > [4] Devlin, Jacob, et al. "Bert: Pre-training of deep bidirectional transformers for language understanding." arXiv preprint arXiv:1810.04805 (2018).
> > ___

---

> > > ### Author Response · Authors · 2025-11-25
> > > **Reviewer fhor - 3/3**
> > >
> > > ___
> > >
> > > ***Q-2: Can you elaborate more on the proposed algorithm and discuss/justify the design choices?***
> > > ___
> > > **QR-2:** Our proposed algorithm, **Pareto-LP-GA**, is designed to solve a specific optimization problem: How do we re-weight training samples to maximize performance on target classes, enforcing performance constraints on others?
> > > We decompose this problem into three distinct design choices:
> > > - **Category-Wise Influence Vectors $P(z)$**
> > >     - **Design:** Instead of a single scalar influence score, we compute a vector $P(z)$ for each sample, where $P_k(z)$ measures the impact of sample $z$ on class $k$.
> > >     - **Justification:** A scalar score hides tradeoffs. As shown in Figure 1, a sample might be beneficial overall (positive scalar) but detrimental to a specific class (negative vector component). The vector formulation is necessary to detect and navigate the "Tradeoff Regions" essential for Pareto analysis.
> > > - **Linear Programming (LP) for Re-weighting:**
> > >     - **Design:** We formulate the re-weighting step as a Linear Program (Algorithm 1, Line 4). We solve for sample weights w that maximize the cumulative influence on target classes $C$,​ subject to the constraint that the influence on non-target classes remains above a threshold $α_k​$.
> > >     - **Justification:** The influence approximation is linear with respect to sample weights. Therefore, finding the optimal weights $w^∗$ to satisfy these multi-objective constraints is a convex optimization problem. LP provides a mathematically guaranteed **global optimum** for the weights given a specific set of constraints [1], which is far more sample-efficient than heuristic data pruning (e.g., top-k removal).
> > > - **Genetic Algorithm (GA) for Constraint Search:**
> > >     - **Design:** We wrap the LP in a Genetic Algorithm to search for the optimal constraint thresholds $α$={$α_1$​,...,$α_K​$}.
> > >     - **Justification:** This is the critical bridge between theory and practice. While influence functions estimate the direction of change, the magnitude of the influence sum does not map linearly to exact accuracy percentages. Since we cannot analytically derive the exact threshold $α_k​$ that corresponds to "zero accuracy drop," we treat α as a hyperparameter to be optimized via GA, using the validation set performance as the fitness function.
> > >
> > > Finally, the LP ensures we are mathematically optimal in the gradient space (influence), while the GA ensures we remain robust in the metric space (accuracy), effectively correcting for the approximation errors inherent in influence functions.
> > >
> > > **References:**\
> > > [1] Boyd, S., & Vandenberghe, L. (2004). Convex Optimization. Cambridge University Press.
> > > ___

---

### Official Review · Reviewer_Lo4Q · 2025-11-03

**Soundness:** 2
**Presentation:** 3
**Contribution:** 2
**Rating:** 4
**Confidence:** 3

**Summary:**

This paper proposes to estimate the influence function at a per class level, instead of the overall influence function on loss in prior literature. The aim is to learn classifier well to reach the pareto fronties. The authors propose an LP based formulation to estimate influence and use it to improve the accuracy in the training. Experiments on CIFAR-10 and Synthetic data are shown for verifying the claims.

**Strengths:**

1.	The paper is clearly written and is intuitively understandable.

2.	With the disclaimer of my limited knowledge in Influence Functions, I find the work to be novel.

**Weaknesses:**

1.	Missing Optimality Guarantees: Although the method is intuitive, it's hard to believe that the proposed algorithm would reach the Pareto frontier without the convergence guarantees for the algorithm.

2.	Missing experiments on Larger Class Datasets: The datasets that are used in the paper are either synthetic or small-scale, like CIFAR-10. It's unclear whether the method can scale to datasets like ImageNet and what its runtime complexity would be.


3.	Hard to parse Fig. 3 and Fig. 4 without detailed captions explaining the results in the figure, as it has a lot of details. Could you please explain how the Spearman correlation is plotted? Is it for validation accuracy?

4.	There are other ways of cost-sensitive learning that could be used to balance the performance of the method across classes.

**Questions:**

1.	In Table 1, the correlation is established for the particular epoch. However the performance at the end of training is more pratical. Could the authors explain how the Influence function improves the final performance?

---

> ### Author Response · Authors · 2025-11-25
> **Reviewer Lo4Q - 1/4**
>
> Dear reviewer Lo4Q, \
> Thank you for your time and effort to review our paper. We really appreciate your observations. Based on your observations we have conducted multiple experiments and we are describing all of those below.
> ___
> ***W-1: Missing Optimality Guarantees: Although the method is intuitive, it's hard to believe that the proposed algorithm would reach the Pareto frontier without the convergence guarantees for the algorithm.***
> ___
> **WR-1:** We acknowledge that deriving strict theoretical convergence guarantees for deep neural networks is challenging due to their non-convex nature. However, we ensure optimality by decoupling the problem into **deterministic stability and empirical verification**. The Linear Programming (LP) solver enforces trajectory stability by executing deterministic and optimal updates, while Geometric Verification provides empirical confirmation of the optimal state, validating that the mathematical conditions required for the Pareto frontier have been physically met.
> - **Global Optimality of the Reweighting Step:** While the global loss landscape is non-convex, our core reweighting step is formulated as a **Linear Programming (LP)** problem. For a fixed set of influence vectors and constraints (determined by the GA), the LP guarantees finding the **globally optimal sample weights** that maximize the influence on target categories [1]. This ensures that each update step is mathematically optimal with respect to the current local approximation.
> - **Geometric Verification of the Frontier:** We do not rely solely on convergence assumptions; we provide a **geometric condition** to verify if the frontier has been reached. As detailed in **Section 3.3 and Figure 1**, the Pareto frontier is achieved when influence vectors align on a tradeoff hyperplane ( y=−x for two classes). We quantitatively measure this using PCA [2]; if the explained variance ratio suggests the vectors do not fit a hyperplane, we know the model has not yet reached the ceiling.
> **References:**\
> [1] Boyd, S., & Vandenberghe, L. (2004). Convex Optimization. Cambridge University Press.\
> [2] Svante Wold, Kim Esbensen, and Paul Geladi. Principal component analysis. Chemometrics and intelligent laboratory systems, 2(1-3):37–52, 1987.
> ___

---

> > ### Author Response · Authors · 2025-11-25
> > **Reviewer Lo4Q - 2/4**
> >
> > ___
> > ***W-2: Missing experiments on Larger Class Datasets: The datasets that are used in the paper are either synthetic or small-scale, like CIFAR-10. It's unclear whether the method can scale to datasets like ImageNet and what its runtime complexity would be.***
> > ___
> > **WR-2:** Thank you for this suggestion. We agree that validating our approach on larger-scale, complex datasets is essential to demonstrate robustness. To address this, we conducted a new set of experiments using a challenging subset of the **ImageNet Dataset**.
> > - **Experimental Setup**
> >     - **Dataset:** ImageNet subset with 55,000 training samples and 11,000 validation samples.
> >     - **Complexity:** Images were resized to 64x64. We have used all the 1k classes, but given the complexity in target class selection and for ease of visualization, the original 1,000 ImageNet classes were mapped to **11 Superclasses** based on the proposed classes in “From ImageNet to Image Classification: Contextualizing Progress on Benchmarks” paper [2].
> >     - **Model Details:** For fair comparison we utilized the same ResNet [3] architecture from our main paper, trained for 20 epochs with a learning rate of 0.005, weight decay of 0.0001, and a training batch size of 512.
> > - **Results and Analysis:** We performed two distinct experiments at the final training stage (Epoch 19) to demonstrate different capabilities of our algorithm. Below we describe the performance improvement in target superclasses. Additionally, we provide a bar plot for better understanding of the changes here: [https://anonymous.4open.science/r/Classifier-Performance-Ceiling-C7FB/examples/imagenet/ImageNet-DI-CC.png](https://anonymous.4open.science/r/Classifier-Performance-Ceiling-C7FB/examples/imagenet/ImageNet-DI-CC.png).
> > | Class | Ep-18 Accuracy (%) | Ep-19 (Weighted) Accuracy (%) | Deviation in Accuracy w.r.t Ep-18 |
> > |-------|----------------------|-------------------------------|------------------------|
> > | C-0   | 49.7 | 57.0 | $\textcolor{green}{+7.3}$ |
> > | C-1   | 43.0 | 37.9 | $-5.1$ |
> > | C-2   | 54.4 | 49.7 | $-4.7$ |
> > | C-3   | 52.1 | 53.7 | $\textcolor{green}{+1.6}$ |
> > | C-4   | 54.6 | 48.1 | $-6.5$ |
> > | C-5   | 41.0 | 40.9 | $-0.1$ |
> > | **C-6**   | 12.9 | 18.6 | $\textcolor{green}{+5.7}$ |
> > | C-7   | 43.1 | 51.5 | $\textcolor{green}{+8.4}$ |
> > | **C-8**   | 31.0 | 32.8 | $\textcolor{green}{+1.8}$ |
> > | **C-9**   | 32.1 | 32.5 | $\textcolor{green}{+0.4}$ |
> > | **C-10**  | 26.7 | 40.4 | $\textcolor{green}{+13.7}$ |
> > | **Average** | **40.1** | **42.1** | $\textcolor{green}{+2.0}$ |
> >
> > - **Direct Improvement:** We targeted superclasses with historically lower performance (Classes 6, 8, 9, and 10) to push the model's capabilities beyond the standard baseline.
> >     - **Result:** The method achieved substantial gains. Notably, **Class 10 saw an improvement of 13.7 in accuracy, and Class-6, 8, and 9 have improvements of 5.7, 1.8, and 0.4** compared to the Epoch 18 baseline. **Overall accuracy is increased by 2.0%**. This demonstrates that our weighting strategy can effectively boost target categories.
> >
> > | ClassWise Acc | Ep-18 Accuracy (%) | Ep-19 Accuracy (%) | Ep-19 (Weighted) Accuracy (%) | Deviation in Accuracy w.r.t Ep-19 |
> > |---------------|---------------------|----------------------|-------------------------------|------------------------|
> > | **C-0**  | 49.7 | 43.6 | 56.7 | $\textcolor{green}{+13.1}$ |
> > | C-1  | 43.0 | 41.9 | 34.3 | $-7.6$ |
> > | **C-2**  | 54.4 | 41.5 | 49.6 | $\textcolor{green}{+8.1}$ |
> > | **C-3**  | 52.1 | 43.0 | 51.1 | $\textcolor{green}{+8.1}$ |
> > | C-4  | 54.6 | 61.4 | 51.6 | $-9.8$ |
> > | C-5  | 41.0 | 43.6 | 37.3 | $-6.3$ |
> > | C-6  | 12.9 | 17.0 | 17.0 | $\textcolor{green}{+0.00}$ |
> > | C-7  | 43.1 | 54.8 | 57.0 | $\textcolor{green}{+2.2}$ |
> > | **C-8**  | 31.0 | 26.4 | 30.6 | $\textcolor{green}{+4.2}$ |
> > | C-9  | 32.1 | 31.1 | 32.1 | $\textcolor{green}{+1.0}$ |
> > | C-10 | 26.7 | 32.7 | 42.7 | $\textcolor{green}{+10.0}$ |
> > | **Average** | **35.3** | **39.7** | **41.8** | $\textcolor{green}{+2.1}$ |
> >
> > - **Course Correction:** We observed that standard training (transitioning from Epoch 18 to 19) caused performance drops in specific classes (Classes 0, 2, 3, and 8). We applied our method to "correct" this regression.
> >     - **Result:** The Pareto-LP-GA-based weighted training successfully reversed the degradation. As shown in the results, **Class 0 shows an improvement of 13.1 in accuracy, and Class-2,3, and 8 get an improvement of 8.1, 8.1 and 4.2** relative to the standard Epoch 19 model. **Additionally, overall accuracy is increased by 2.1**. This confirms the method's ability to recover lost performance in specific categories without retraining from scratch.
> >
> > **[Continuing WR-2] ...**

---

> ### Author Response · Authors · 2025-11-25
> **Reviewer Lo4Q - 3/4**
>
> **...**
>
> **Scalability Analysis:** Our algorithm scales linearly with dataset size, making it feasible for large benchmarks like ImageNet.
> - **Outer Loops ($G⋅|P$|):** We run G generations with a population of size $|P|$.
> - **Data Loop (|$T$|):** For every candidate, we process the full training set $|T|$.
> - **Inner Calculation ($M+K^{1.5}$):**
>     - **Deep Learning Inference ($M$):** Running the neural network.
>     - **Optimization Solving ($K^{1.5}$):** Solving the linear program. [4,5]
>     - **Overall complexity ≈ $\mathcal{O}$($G⋅|P|⋅|T|⋅[M+K^{1.5}]$)**, where, $G$ = Generations, $P$ = Population Size, $T$ = Size of Training Set, $M$ = Time complexity of Deep Learning Model, $K$ = Number of Classes
>
> **References:**\
> [1] J. Deng, W. Dong, R. Socher, L. -J. Li, Kai Li and Li Fei-Fei, "ImageNet: A large-scale hierarchical image database," CVPR 2009.\
> [2] Tsipras et. al., “From ImageNet to Image Classification: Contextualizing Progress on Benchmarks”. ICML 2020.\
> [3] He, Kaiming, et al. "Deep residual learning for image recognition." CVPR 2016.\
> [4] Lee, Y. T., & Sidford, A. (2014). “Path Finding Methods for Linear Programming: Solving Linear Programs in O(√rank) Iterations and Faster Algorithms for Maximum Flow.”\
> [5] Lee, Y. T., & Sidford, A. (2015). Efficient Inverse Maintenance and Faster Algorithms for Linear Programming.
> ___
>
> ___
>
> ***W-3: Hard to parse Fig. 3 and Fig. 4 without detailed captions explaining the results in the figure, as it has a lot of details. Could you please explain how the Spearman correlation is plotted? Is it for validation accuracy?***
> ___
> **WR-3:** We apologize for the lack of clarity and will improve the figure by adding detailed captions. Below, we provide additional clarifications regarding the figures. The scatter plots (Subfigures C and F) are derived directly from the heatmaps (A/B and D/E).
> - **Data Points:** Each point represents a specific pair of classes. For example, if we remove samples beneficial to Class 0 (Target), we measure the resulting influence and accuracy change on Class 0, Class 1,..., Class $K$-1. This generates K data points for that single removal experiment. We conduct one experiment for removing beneficial samples (A, B) and one for removing detrimental samples (D, E).
> - **Axes for figure A and D:**
>     - **X-axis:** The cumulative **category-wise influence score** on a specific class when we remove beneficial (A) or detrimental (D) samples of a target class.
>     - **Y-axis:** Targeted class for removing beneficial (A) or detrimental (D) samples.
> - **Axes for figure B and E:**
>     - **X-axis:** The cumulative **category-wise change in accuracy** on a specific class when we remove beneficial (B) or detrimental (E) samples of a target class.
>     - **Y-axis:** Targeted class for removing beneficial (A) or detrimental (D) samples.
>     - **Spearman Correlation:** This is calculated across all these aggregated data points to quantify how well our influence metric ranks the actual performance changes across all categories. **A high correlation (>0.8) confirms that if the influence vector predicts a large drop for a specific class, that drop actually occurs.**
>
> **Validation Accuracy:** Yes, the accuracy plotted is **Validation Accuracy**.
> - As defined in Equation (1) and Section 3.1, our influence vectors are estimated using the gradient and Hessian computed on the **validation set** (to approximate generalization error).
> - Consequently, the "Performance Change" is measured by retraining the model and evaluating it on the same held-out validation set to ensure consistency with the influence estimation objective.
>
> We will add these points in our manuscript for better clarity and upload a new version within a week.
> ___

---

> > ### Author Response · Authors · 2025-11-25
> > **Reviewer Lo4Q - 4/4**
> >
> > ___
> >
> > ***W-4: There are other ways of cost-sensitive learning that could be used to balance the performance of the method across classes.***
> > ___
> > **WR-4:** Thank you for sharing your observation on cost-sensitive learning. However, the fundamental difference lies in the **research question, granularity of control and the optimization mechanism**.
> > - **Research Question:** The target research question of our paper aims to answer whether a model achieves its performance ceiling in the multi-class scenario; while the cost-sensitive learning pre-defines the tradeoff importance among mutli-class as a new objective function for optimization.
> >
> > - **Sample-Level vs. Class-Level Granularity:** Standard cost-sensitive learning assigns a static weight $λ_k$​ to an entire class $k$. This assumes all samples within a class contribute equally to the objective. In contrast, our method assigns a specific weight $w_i$​ to each training sample $z_i$​ based on its unique influence vector $P(z_i​)$. This allows us to upweight a specific sample that benefits multiple target classes while downweighting a sample from the same class that creates conflicts.
> >
> > - **Handling Tradeoffs via Geometry:** Cost-sensitive methods cannot distinguish between "Joint Positive" samples (beneficial to all) and "Tradeoff" samples (beneficial to one, harmful to another) within the same class. Our category-wise influence vectors explicitly map these regions (**Figure 1**), enabling the Linear Program (LP) to mathematically select the optimal combination of samples that resolves these conflicts.
> > ___
> > ___
> >
> > ***Q-1: In Table 1, the correlation is established for the particular epoch. However, the performance at the end of training is more practical. Could the authors explain how the Influence function improves the final performance?***
> > ___
> > **QR-1:** We agree that final performance is the most practical metric. To demonstrate this, we applied our **Pareto-LP-GA** method at the final stage of training (Epoch 19) on CIFAR-10. The results confirm that our influence-based reweighting functions as an effective "final polish" to optimize the Pareto frontier of the deployed model.
> > We conducted two sets of experiments at the final epoch:
> > | ClassWise Acc | Ep-18 | Ep-19 (Weighted) | (%) Deviation w.r.t Ep-18 |
> > |---------------|--------|-------------------|------------------------|
> > | C-0 | 0.904 | 0.895 | $-1.00\%$ |
> > | C-1 | 0.838 | 0.868 | $\textcolor{green}{+3.5\%}$ |
> > | **C-2** | 0.703 | 0.704 | $\textcolor{green}{+0.14\%}$ |
> > | **C-3** | 0.724 | 0.733 | $\textcolor{green}{+1.24\%}$ |
> > | **C-4** | 0.781 | 0.810 | $\textcolor{green}{+3.71\%}$ |
> > | **C-5** | 0.758 | 0.795 | $\textcolor{green}{+4.88\%}$ |
> > | C-6 | 0.869 | 0.879 | $\textcolor{green}{+1.15\%}$ |
> > | C-7 | 0.832 | 0.819 | $-1.56\%$ |
> > | C-8 | 0.918 | 0.919 | $\textcolor{green}{+0.11\%}$ |
> > | C-9 | 0.881 | 0.898 | $\textcolor{green}{+1.92\%}$ |
> >
> > - **Direct Improvement:** We targeted classes with lower performance (Classes 2, 3, 4, 5) to push the model's final ceiling.
> >     - **Result:** As shown in the table below, we achieved significant gains in the target categories compared to the baseline Epoch 18, with **Class 5 improving by +4.88% and Class 4 by +3.71%**, while maintaining stability in other high-performing classes (Class 6: +1.15%, Class 9: +1.92%).
> >
> >
> > - **Course Correction:** We observed that standard training (Epoch 19) caused a performance regression in **Class 3** and **Class 7** compared to Epoch 18. We applied our method to "correct" this deviation.
> > Result: Our weighted training successfully prevented this degradation. The weighted model outperformed the standard Epoch 19 model by **+0.42% on Class 3 and +0.48% on Class 7**, effectively steering the final model back to a better Pareto trade-off.
> >
> > | ClassWise Acc | Ep-18 | Ep-19 | Ep-19 (Weighted) (Target = 3,7) | Deviation w.r.t Ep-19 |
> > |---------------|-------|-------|----------------------------------|------------------------|
> > | C-0 | 0.904 | 0.897 | 0.897 | $\textcolor{green}{+0.00\%}$ |
> > | C-1 | 0.838 | 0.871 | 0.863 | $-0.92\%$ |
> > | C-2 | 0.703 | 0.725 | 0.711 | $-1.93\%$ |
> > | **C-3** | 0.724 | 0.700 | 0.703 | $\textcolor{green}{+0.42\%}$ |
> > | C-4 | 0.781 | 0.786 | 0.784 | $-0.25\%$ |
> > | C-5 | 0.758 | 0.810 | 0.814 | $\textcolor{green}{+0.49\%}$ |
> > | C-6 | 0.869 | 0.872 | 0.879 | $\textcolor{green}{+0.80\%}$ |
> > | **C-7** | 0.832 | 0.827 | 0.831 | $\textcolor{green}{+0.48\%}$ |
> > | C-8 | 0.918 | 0.923 | 0.920 | $-0.32\%$ |
> > | C-9 | 0.881 | 0.892 | 0.905 | $\textcolor{green}{+1.56\%}$ |
> >
> > These results demonstrate that our method is not just for intermediate diagnostics but is a powerful tool for **final model refinement**, allowing developers to dictate the exact performance profile of the deployed classifier.
> >
> > ___

---

### Meta-Review · Area_Chair_Jvbg · 2026-01-04

**Summary:**

The paper presents a clearly written and intuitive approach, and several reviewers acknowledge the simplicity and potential usefulness of the formulation. However, the majority of reviewers raise substantial concerns regarding novelty, theoretical validity, and positioning with respect to prior work. The core idea of class-wise or vectorized influence functions is viewed as incremental, with limited new insight beyond existing influence-based and data-centric methods. Crucially, comparisons with closely related literature (e.g., FairIF, D3M, Chhabra et al., and connections to hard negative mining) are either missing or insufficiently addressed, weakening the empirical and conceptual contribution.

**Reviewer Concerns:**

The addition of ImageNet-scale experiments (≈55k samples) directly addresses the concern that the method was only validated on synthetic or small datasets like CIFAR-10. But using 55k samples is still close to CIFAR kind of smaller dataset.

While the rebuttal argues that vectorized influence functions and Pareto diagnostics are new, reviewers’ concerns that the method relies on first-order influence approximations without deeper theoretical insight are not fully resolved.

[fhor, e15q]
Some clarifications regarding Algorithmic clarity and design justification were added, but reviewers’ confusion around the fitness function, GA design choices, and Pareto interpretation suggests that these explanations may still be insufficient

**Reviewer Scores:**

Most of the reviewers would have retained their scores and would not have crossed the borderline reject score, even after the increase.

---

### Decision · Program_Chairs · 2026-01-26

Reject